# Mechanistic models of Rift Valley fever virus transmission: A systematic review

**Hélène Cecilia**[1]☉¤*, **Alex Drouin**[2,3]☉, **Raphaëlle Métras**[4,5], **Thomas Balenghien**[2,6,7], **Benoit Durand**[3]*, **Véronique Chevalier**[2,8,9], **Pauline Ezanno**[1]

**1** Oniris, INRAE, BIOEPAR, Nantes, France, **2** ASTRE, University of Montpellier, CIRAD, INRAE, Montpellier, France, **3** Université Paris-Est, Anses, Laboratory for Animal Health, Epidemiology Unit, Maisons-Alfort, France, **4** Sorbonne Université, INSERM, Institut Pierre Louis d'Épidémiologie et de Santé Publique (IPLESP, UMRS 1136), Paris, France, **5** Department of Infectious Disease Epidemiology, Centre for the Mathematical Modelling of Infectious Diseases, London School of Hygiene and Tropical Medicine, London, United Kingdom, **6** CIRAD, UMR ASTRE, Rabat, Morocco, **7** IAV Hassan II, UR MIMC, Rabat, Morocco, **8** CIRAD, UMR ASTRE, Antananarivo, Madagascar, **9** Institut Pasteur de Madagascar, Epidemiology and Clinical Research Unit, Antananarivo, Madagascar

☉ These authors contributed equally to this work.
¤ Current address: Department of Biology, New Mexico State University, Las Cruces, New Mexico, United States of America
* helene.cecilia3@gmail.com (HC); benoit.durand@anses.fr (BD)

**Data Availability Statement:** All relevant data are within the manuscript and its Supporting Information files.

## Abstract

Rift Valley fever (RVF) is a zoonotic arbovirosis which has been reported across Africa including the northernmost edge, South West Indian Ocean islands, and the Arabian Peninsula. The virus is responsible for high abortion rates and mortality in young ruminants, with economic impacts in affected countries. To date, RVF epidemiological mechanisms are not fully understood, due to the multiplicity of implicated vertebrate hosts, vectors, and ecosystems. In this context, mathematical models are useful tools to develop our understanding of complex systems, and mechanistic models are particularly suited to data-scarce settings. Here, we performed a systematic review of mechanistic models studying RVF, to explore their diversity and their contribution to the understanding of this disease epidemiology. Researching Pubmed and Scopus databases (October 2021), we eventually selected 48 papers, presenting overall 49 different models with numerical application to RVF. We categorized models as theoretical, applied, or grey, depending on whether they represented a specific geographical context or not, and whether they relied on an extensive use of data. We discussed their contributions to the understanding of RVF epidemiology, and highlighted that theoretical and applied models are used differently yet meet common objectives. Through the examination of model features, we identified research questions left unexplored across scales, such as the role of animal mobility, as well as the relative contributions of host and vector species to transmission. Importantly, we noted a substantial lack of justification when choosing a functional form for the force of infection. Overall, we showed a great diversity in RVF models, leading to important progress in our comprehension of epidemiological mechanisms. To go further, data gaps must be filled, and modelers need to improve their code accessibility.

**Funding:** This work was part of the FORESEE project funded by INRAE metaprogram GISA (Integrated Management of Animal Health). HC was funded by INRAE, Région Pays de la Loire, CIRAD. We also would like to acknowledge the support of the French Ministry of Agriculture, which funded this research (AD). The funders had no role in study design, data collection and analysis, decision to publish, or preparation of the manuscript.

**Competing interests:** The authors have declared that no competing interests exist.

## Author summary

Rift Valley fever (RVF) affects humans and livestock across Africa, South West Indian Ocean islands, and the Arabian Peninsula. This disease is one of the World Health Organization priorities and is caused by a virus which is transmitted by mosquitoes (mainly of *Aedes* and *Culex* spp. genera), but also by direct contact from livestock to humans. Mathematical models have been used in the last 20 years to disentangle RVF virus transmission dynamics. These models can further improve our understanding of processes driving outbreaks, test the efficiency of control strategies, or even anticipate possible emergence. Provided with detailed datasets, models can tailor their conclusions to specific geographical contexts and aid in decision-making in the field. This review provides a general overview of mathematical models developed to study RVF virus transmission dynamics. We describe their main results and methodological choices, and identify hurdles to be lifted. To offer innovative animal and public health value, we recommend that future models focus on the relative contribution of host and vector species to transmission, and the role of animal mobility.

## Introduction

Rift Valley fever (RVF) is a viral, vector-borne, zoonotic disease, first identified in Kenya in 1930 [1]. It has since then been reported across the African continent, in the South West Indian Ocean islands, and in the Arabian Peninsula. Transmission of Rift Valley fever virus (RVFV) mainly involves *Aedes* and *Culex* spp. mosquitoes [2], some of which are present in Europe and North America [3–10], but other genera may also be potential vectors [11–14]. In livestock, abortion storms and death can strongly impact the local economy [15,16]. Human infections arise mostly following contact with tissues of infected animals but is also vector-mediated. The clinical spectrum in humans is broad, with a minority of deadly cases [17,18].

About 100 years after its first description, RVF outbreaks are still difficult to anticipate and contain, and the drivers of RVF endemicity are not clearly understood. The multiplicity of vertebrate host and mosquito species involved, the diversity of affected ecosystems, each with their own environmental dynamics, as well as the impact of human activities, make this complex system hard to disentangle [19]. The limited use of available vaccines [20], coupled with the overall social vulnerability of affected regions [21,22], are also major obstacles. The pastoralist tradition, which constitutes the main production system in African drylands [23], can induce delayed access to health care and hinder the traceability of animal mobility. This, in turn, impacts the quality and the availability of epidemiological data, which can be quite heterogeneous [24–26]. As a result, it is often difficult to generalize local findings, unless a mechanistic understanding of epidemiological processes is acquired.

Mathematical models are useful to project epidemiological scenarios, including control strategies. This can be done at large scales (temporal [27], spatial [28], or demographic [28]). Powerful methods can now estimate the most likely drivers of observed outbreak patterns [29,30], or point out key processes needing further field or laboratory investigations [31]. Phenomenological models, be they mathematical or statistical, aim at extracting patterns and information from data, with no focus on underlying mechanisms responsible for such observed patterns [32]. By contrast, mechanistic (sometimes called dynamical) models explicitly include processes governing the system of interest [32]. Consequently, mechanistic models can adapt to data-scarce settings by exploring a complex system conceptually, in a hypothesis-driven fashion [33], e.g., to see what ranges of behavior can emerge from first principles, as is routinely done in ecology [34]. This flexibility gives rise to an interesting variability in the way

epidemiological mechanistic models are designed and used, spanning a broad spectrum from highly theoretical to closely mimicking field situations [35].

Two existing reviews have focused on models developed to study RVF. The first one, by Métras et al. (2011) [36], was a narrative review presenting modeling tools used to measure or model the risk of RVF occurrence in animals. At that time, only three mechanistic models were available and included in the study. The second one, by Danzetta et al. (2016) [37], was a systematic review constrained to compartmental models which included 24 articles. The authors used RVF as a case study to present how the use of compartmental models can be helpful to investigate various aspects of vector-borne disease transmission. A complementary paper, by Reiner et al. (2013) [38], reviewed 40 years of mathematical models of mosquito-borne pathogen transmission, with a thorough and comprehensive reading grid. It did however only include three models on RVF.

To update the state-of-the-art on mechanistic models of RVFV transmission, we conducted a systematic review. Our main goal was to identify knowledge gaps left unaddressed by models, and therefore identify future research avenues. To achieve this, we categorized models on a spectrum from theoretical to applied (the middle-ground category being called 'grey') and explored these categories throughout the paper to identify what they have in common and how they differ. First, we explored their inheritance connections and assessed whether these categories inspired each other. We then detailed their contribution to the understanding of RVF epidemiology. Lastly, we described the diversity of methodological choices and assumptions made in these models. In particular, we dedicated a whole section to present the different functional forms used by models for the force of infection. We detailed the underlying assumptions on host-vector interactions that these functional forms imply, as we noticed a lack of justification regarding this choice in reviewed papers, even though host-vector interactions represent a key factor in RVFV transmission. In that regard, we therefore insist that key results presented in this review should be interpreted with this methodological choice in mind.

## Material and methods

### Search strategy

This review was conducted according to the Preferred Reporting Items for Systematic reviews and Meta-Analyses (PRISMA) guidelines [39,40]. The research was performed in Scopus and Pubmed databases on 12 October 2021. No restriction on publication date was considered. The following Boolean query was applied in both databases: (*rift* AND *valley* AND *fever*) AND (*mathematical* OR *epidem*\* OR *compartment*\* OR *sir* OR *seir* OR *metapopulation* OR *deterministic* OR *stochastic* OR *mechanistic* OR *dynamic*\*) AND (*model*\*).

This query was used in the "title, abstract, and keywords", and "title and abstract" fields for Scopus and PubMed, respectively.

### Inclusion and exclusion criteria

After removal of duplicates, studies were included in three steps: title screening, abstract screening, and full text reading. In the first and second steps, records were selected if they appeared to present a RVF model using a mechanistic approach for at least one part of the model. Exclusion criteria were: irrelevant topic, reviews, case reports, serological studies, and statistical studies. Records selected in the first and second steps went to a full text screening of the corresponding report, using a combination of the first set of exclusion criteria along with the following additional ones: non-mechanistic models, models representing mosquitoes only, incomplete model description, and theoretical papers without any RVF numerical application. Discussion among authors occurred in case of doubt to reach a consensus on final inclusions.

### Screening

We designed a reading grid (S1 Text), partially inspired by the one used in Reiner et al. (2013) [38], to collect information from the studies. The context of the study (e.g., location, presence of data), model components (e.g., host and vector species, infection states), and assumptions (e.g., vertical transmission in vectors), type of outputs (e.g., $R_0$, parameter estimations, sensitivity analysis), and main results were all recorded. Two authors took charge of the systematic reading. To cross-validate the use of the grid, three studies were read by both authors and specific topics were regularly discussed to make sure a consensus was reached.

### Model typology and inheritance connections

We defined three model categories: theoretical, applied, and grey models. Theoretical models do not use any data and are not intended to represent any specific geographical location. Applied models represent a specific geographical context and use relevant data to tailor model development to their case study or to validate model outputs. Such data can be of several types, as environmental or demographic data, and not necessarily epidemiological in the sense of seroprevalence or case reports. Grey models are those which do not fit into these well-defined categories. In some cases, authors do not use data but demonstrate a strong will to adapt their models to a specific geographical or epidemiological context. In other cases, despite the use of data, the model developed is still very conceptual and lacks realism in its key features. In such cases, the model analysis rarely deepens the epidemiological understanding of the pathosystem. We recorded inheritance connections between studies: if a model stated being adapted from another model, we defined the latter as a parent model.

## Results and discussion

### Study selection

A total of 372 records were identified from the two databases. After removal of duplicates, 248 records were screened at the title level, 146 at the abstract level, and 69 reports were fully read. Twenty-one reports were excluded during full-text reading: three were excluded due to incomplete model description [41–43], three modeled mosquito population only [44–46], ten were not mechanistic models [47–56], three were review papers [57–59] and two were theoretical without application to RVF [60,61]. Eventually, 49 studies were selected for the present review (Fig 1). Among those, 26 were not present in the review by Danzetta et al. (2016) [37].

### Model typology and inheritance connections

We identified 18 applied models (37%), 18 theoretical models (37%), and 13 grey models (26%, Table 1). Twenty-one models (43%) had a parent model within the list of presently reviewed studies, for a total of twenty-seven models in the inspirational network (Fig 2). In 15 cases (71%), a model and its parent shared at least one author. In 14 cases, a models and its parent belonged to the same category (6 applied, 1 grey, 7 theoretical). The model by Gaff et al. (2007) [62] is a clear example of a model laying the groundwork for future model developments. It was first modified to explore several control strategies in Gaff et al. (2011) [63] (theoretical). Adongo et al. (2013) [64] (theoretical) then elaborated on Gaff et al. (2011) [63] to explore sophisticated vaccination schemes. Besides, Gaff et al. (2007) [62] model was spatialized in Niu et al. (2012) [65] (theoretical). In other cases, theoretical and grey studies provided a basis for the construction of more applied models in further work. One grey model [66] was the parent of an applied model [67]. In four cases, a theoretical model ([62] twice, [68], [69])

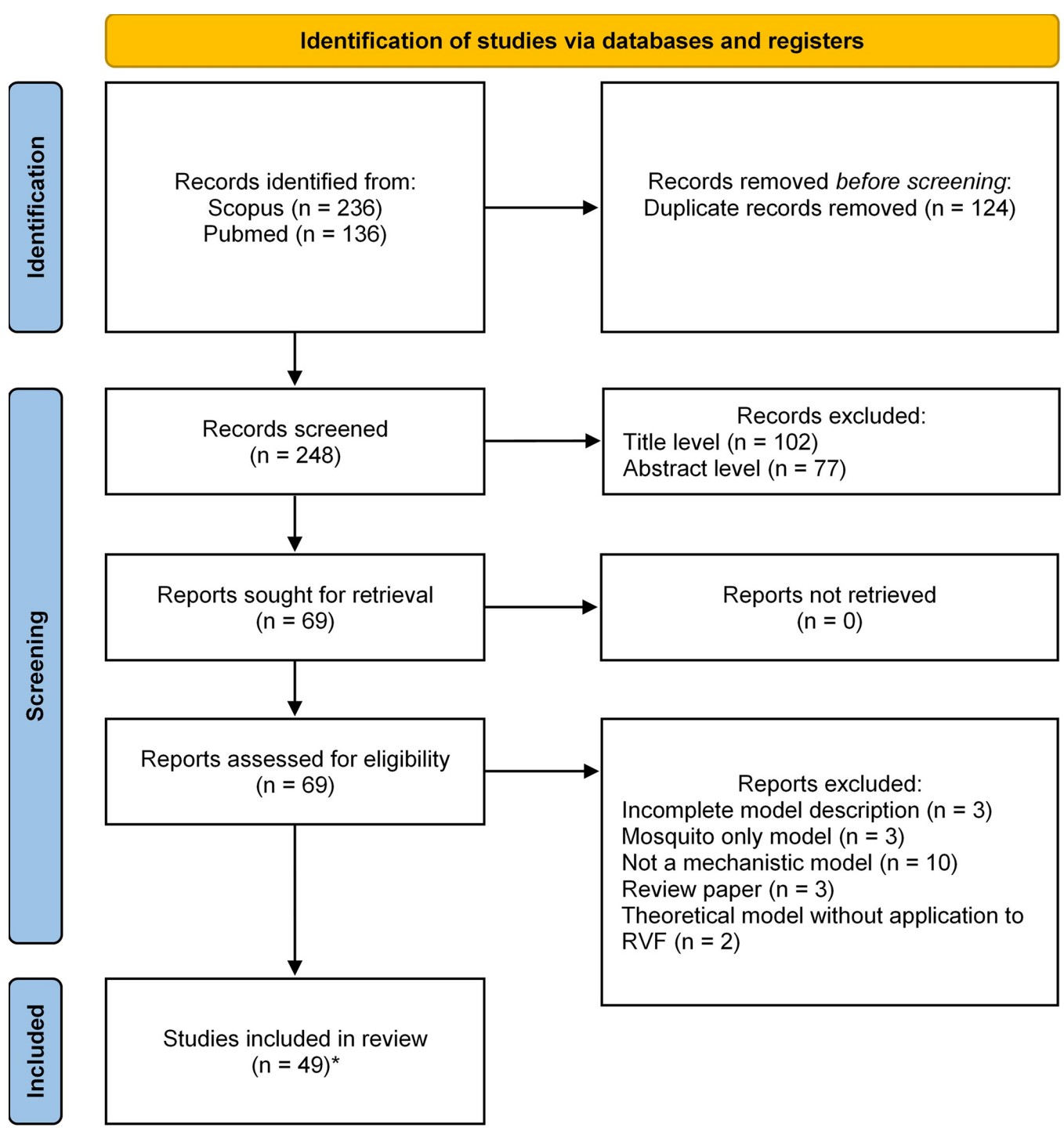

**Fig 1. PRISMA flow diagram representing the selection process.** Record: title and/or abstract of a report indexed in a database. Report: document supplying information about a study. Study: An experiment, corresponding here to models [39]. * One report included two studies.

was a parent of a grey model ([70], [71], [72], and [66] respectively). Lastly, Gaff et al. (2007) [62], a theoretical model, was the parent of two applied models [73,74].

Changes in model features can also give an overview of the continuity between a model and its parent. Métras et al. (2020) [77] added a human compartment to the model of

**Table 1. Main characteristics of mechanistic models on Rift Valley fever virus transmission included in the review.**

| Study | Model category | Primary objective | Main output | Deterministic or stochastic? | Geographical zone | Scale | Number vertebrate hosts taxa | Taxa | Host infection states | Number vector taxa | Taxa | Vector infection states | FOI | Compartmental or ABM? | Code access |
|---|---|---|---|---|---|---|---|---|---|---|---|---|---|---|---|
| Beechler et al. (2015) [67] | Applied | Understand | Scenario comparison | Deterministic | South Africa | Local | 1 | Buffalo | SIR | 1 | Aedes (assumed) | SEI | Hybrid1 | Compartmental | No |
| Bicout and Sabatier (2004) [85] | Applied | Understand | Scenario comparison | Deterministic with at least one stochastic process | Senegal | Local | 1 | Livestock | IR (other states not described) | 2 | Aedes, Culex | Not explicit | FR | Compartmental | No |
| Scoglio et al. (2016) [86] | Applied | Understand | Scenario comparison | Stochastic | United States of America | Sub-national | 1 | Cattle | SEIR | 0 | | | NA | ABM | Yes |
| Sekamatte et al. (2019) [87] | Applied | Understand | Scenario comparison | Stochastic | Uganda | Sub-national | 1 | Cattle | SEIR | 0 | | | NA | ABM | No |
| Leedale et al. (2016) [82] | Applied | Understand | Risk map | Deterministic | Kenya, Tanzania | International | 1 | Livestock | SEIR | 2 | Aedes, Culex | SEI | FR | Compartmental | No |
| Cecilia et al. (2020) [88] | Applied | Understand | Risk map | Deterministic | Senegal | Sub-national | 2 | Cattle, Small ruminants | SEIR | 2 | Aedes, Culex | SEI | FR | Compartmental | Yes |
| Xue et al. (2013) [76] | Applied | Understand | Risk map | Deterministic with at least one stochastic process | United States of America | Sub-national | 2 | Cattle, Human | SEIR | 2 | Aedes, Culex | SEI | FI | Compartmental | No |
| Xue et al. (2012) [74] | Applied | Understand | Parameter estimation | Deterministic | South Africa | Sub-national | 2 | Sheep, Human | SEIR | 2 | Aedes, Culex | SEI | FI | Compartmental | No |
| Nicolas et al. (2014) [90] | Applied | Understand | Parameter estimation | Deterministic | Madagascar | Sub-national | 1 | Cattle | SEIR | 1 | Not precised | SEI | FR | Compartmental | No (upon request) |
| Métras et al. (2017) [78] | Applied | Understand | Parameter estimation | Deterministic | Mayotte | Sub-national | 1 | Livestock | SEIR | 0 | | | NA | Compartmental | No |
| Métras et al. (2020) [77] | Applied | Understand | Parameter estimation | Deterministic | Mayotte | Sub-national | 2 | Livestock, Human | SEIRVsVp | 0 | | | NA | Compartmental | Yes |
| Tennant et al. (2021) [80] | Applied | Understand | Parameter estimation | Deterministic | Comoros archipelago | International | 1 | Livestock | SEIR | 0 | | | NA | Compartmental | Yes |
| Durand et al. (2020) [91] | Applied | Understand | Parameter estimation | Deterministic with at least one stochastic process | Senegal | Local | 2 | Cattle, Small ruminants | SIR | 2 | Aedes, Culex | SEI | FR | Compartmental | No |
| Barker et al. (2013) [73] | Applied | Anticipate | Risk map | Deterministic | United States of America | Sub-national | 2 | Cattle, Birds | SEIR | 2 | Aedes, Culex | SEI | FI | Compartmental | No |
| Fischer et al. (2013) [92] | Applied | Anticipate | Risk map | Deterministic | Netherlands | National | 2 | Cattle, Small ruminants | SEIR | 2 | Aedes, Culex | SEI | FR | Compartmental | No |
| Taylor et al. (2016) [81] | Applied | Anticipate | Risk map | Deterministic | East African Community | International | 1 | Livestock | R (other states not described) | 2 | Aedes, Culex | Not specified | FR | Compartmental | No |
| EFSA AHAW Panel et al. (2020 – Model 1) [79] | Applied | Control | Scenario comparison | Stochastic | Mayotte | Sub-national | 1 | Livestock | SEIRV | 1 | Culex | SEI | NA | Compartmental | No |

*(Continued)*

**Table 1.** (Continued)

| Study | Model category | Primary objective | Main output | Deterministic or stochastic? | Geographical zone | Scale | Number vertebrate hosts taxa | Taxa | Host infection states | Number vector taxa | Taxa | Vector infection states | FOI | Compartmental or ABM? | Code access |
|---|---|---|---|---|---|---|---|---|---|---|---|---|---|---|---|
| EFSA AHAW Panel et al. (2020 – Model 2) [79] | Applied | Control | Risk map | Stochastic | Netherlands | National | 1 | Livestock | SIR | 0 | | | FR | Compartmental | No |
| Gao et al. (2013) [84] | Grey | Understand | Scenario comparison | Deterministic | Sudan, Egypt | International | 1 | Livestock | SIR | 1 | Not precised | SI | MA | Compartmental | No |
| Manore and Beechler (2015) [66] | Grey | Understand | Scenario comparison | Deterministic | South Africa | Local | 1 | Buffalo | SIR | 1 | Aedes | SEI | Hybrid1 | Compartmental | No |
| Xiao et al. (2015) [83] | Grey | Understand | Scenario comparison | Deterministic | Sudan, Egypt | International | 1 | Livestock | SEIR | 1 | Culex | SEI | MA | Compartmental | No |
| Lo Iacono et al. (2018) [93] | Grey | Understand | Scenario comparison | Deterministic | Kenya | National | 1 | Livestock | SEIR | 2 | Aedes, Culex | SEI | Hybrid3 | Compartmental | No |
| Sumaye et al. (2019) [94] | Grey | Understand | Scenario comparison | Deterministic | Tanzania | Sub-national | 2 | Cattle, Human | SEIR | 4 | Aedes, Culex | SI | Hybrid1 | Compartmental | Yes |
| McMahon et al. (2014) [95] | Grey | Understand | Scenario comparison | Deterministic with at least one stochastic process | East Africa | International | 3 | Cattle, Wildlife, Human | SEIRAV | 2 | Aedes, Culex | SEI | Hybrid2 | Compartmental | No |
| Gil et al. (2016) [96] | Grey | Understand | Scenario comparison | Both tested | Egypt | National | 1 | Livestock | SIR | 1 | Culex | SI | MA | Both tested | No |
| Tuncer et al. (2016) [97] | Grey | Understand | Parameter estimation | Deterministic | Kenya | Sub-national | 2 | Livestock, Human | SI-R for livestock only | 1 | Not precised | SI | MA* | Compartmental | No |
| Cavalerie et al. (2015) [71] | Grey | Understand | Parameter estimation | Stochastic | Mayotte | Sub-national | 1 | Livestock | SEIR | 1 | Mean from several species | SEI | FR | Compartmental | Yes |
| Mpeshe et al. (2014) [72] | Grey | Understand | Sensitivity analysis | Deterministic | Tanzania | Sub-national | 2 | Livestock, Human | SEIR | 2 | Aedes, Culex | SEI | FI | Compartmental | No |
| Pedro et al. (2016) [98] | Grey | Understand | Mathematical properties | Stochastic | East Africa, South Africa | | 1 | Livestock | SIR | 1 | Aedes | SI | Hybrid1 | ABM | No |
| Miron et al. (2016) [70] | Grey | Anticipate | Sensitivity analysis | Deterministic | North America | Local | 2 | Livestock, Human | SEIR | 1 | Aedes | SEI | MA | Compartmental | No |
| Gachohi et al. (2016) [99] | Grey | Control | Scenario comparison | Deterministic | Kenya | Local | 2 | Cattle, Small ruminants | SEIR | 2 | Aedes, Culex | SEI | FR | Compartmental | Yes |
| Niu et al. (2012) [65] | Theoretical | Understand | Scenario comparison | Deterministic | | | 1 | Livestock | SEIR | 2 | Aedes, Culex | SEI | FI | Compartmental | No |
| Chamchod et al. (2014) [100] | Theoretical | Understand | Scenario comparison | Deterministic | | | 1 | Livestock | SIR | 1 | Not precised | SI | FR | Compartmental | No |
| Pedro (2018) [101] | Theoretical | Understand | Scenario comparison | Deterministic | | | 1 | Livestock | SIR | 1 | Aedes | SI | FR | Compartmental | No |
| Wen et al. (2019) [102] | Theoretical | Understand | Scenario comparison | Deterministic | | | 1 | Livestock | SEIR | 1 | Not precised | SEI | MA | Compartmental | No |
| Python Ndekou Tandong et al. (2020) [89] | Theoretical | Understand | Scenario comparison | Deterministic | | | 1 | Animals | SEIR | 2 | Aedes, Culex | SEI | FI | Mixed** | No |

(Continued)

**Table 1.** (Continued)

| Study | Model category | Primary objective | Main output | Deterministic or stochastic? | Geographical zone | Scale | Number vertebrate hosts taxa | Taxa | Host infection states | Number vector taxa | Taxa | Vector infection states | FOI | Compartmental or ABM? | Code access |
|---|---|---|---|---|---|---|---|---|---|---|---|---|---|---|---|
| Mpeshe (2021) [103] | Theoretical | Understand | Scenario comparison | Deterministic | | | 1 | Human | SEIR | 1 | Not precised | SEI | FI | Compartmental | No |
| Xue and Scoglio (2015) [104] | Theoretical | Understand | Scenario comparison | Deterministic with at least one stochastic process | | | 1 | Livestock | SEIR | 1 | Not precised | SEI | FR | Compartmental | No |
| Gaff et al. (2007) [62] | Theoretical | Understand | Sensitivity analysis | Deterministic | | | 1 | Livestock | SEIR | 2 | Aedes, Culex | SEI | FI | Compartmental | No |
| Mpeshe et al. (2011) [68] | Theoretical | Understand | Sensitivity analysis | Deterministic | | | 2 | Livestock, Human | SEIR | 1 | Not precised | SEI | FI | Compartmental | No |
| Chitnis et al. (2013) [69] | Theoretical | Understand | Sensitivity analysis | Deterministic | | | 1 | Cattle | SIR | 1 | Aedes | SEI | Hybrid1 | Compartmental | No |
| Xue and Scoglio (2013) [105] | Theoretical | Understand | Sensitivity analysis | Deterministic | | | 2 | Livestock, Human | SEIR | 2 | Aedes, Culex | SEI | FI | Compartmental | No |
| Pedro et al. (2016) [75] | Theoretical | Understand | Sensitivity analysis | Deterministic | | | 1 | Livestock | SIRA | 2 | Aedes, Culex | SEI | Hybrid1 | Compartmental | No |
| Pedro et al. (2017) [106] | Theoretical | Understand | Sensitivity analysis | Deterministic | | | 1 | Livestock | SIR | 3 | Aedes, Culex, Hyalomma ticks | SE (Mosquitoes only) I | Hybrid1 | Compartmental | No |
| Pedro et al. (2014) [107] | Theoretical | Understand | Mathematical properties | Deterministic | | | 1 | Livestock | SIRA | 2 | Aedes, Culex | SEI | Hybrid1 | Compartmental | No |
| Gaff et al. (2011) [63] | Theoretical | Control | Scenario comparison | Deterministic | | | 1 | Cattle | SEIRV | 2 | Aedes, Culex | SEI | FI | Compartmental | No |
| Adongo et al. (2013) [64] | Theoretical | Control | Scenario comparison | Deterministic | | | 1 | Livestock | SEIR | 2 | Aedes, Culex | SEI | FI | Compartmental | No |
| Chamchod et al. (2016) [108] | Theoretical | Control | Scenario comparison | Deterministic | | | 1 | Livestock | SIRV1V2 | 1 | Not precised | SI | MA | Compartmental | No |
| Yang and Nie (2016) [109] | Theoretical | Control | Scenario comparison | Deterministic | | | 1 | Livestock | SIR | 1 | Not precised | SI | MA | Compartmental | No |

We chose not to assign a scale to theoretical models, as well as those with a vaguely defined geographical context. Note that computations covered timespans from 2 months to tens of years. Meaning of abbreviated infection states: susceptible (S), exposed (E), infected (I), recovered (R), asymptomatic (A), Vaccinated but still susceptible (Vs), Vaccinated and protected (Vp), Vaccinated (V), vaccinated by live vaccines (V1), vaccinated by killed vaccines (V2). FOI: force of infection (functional form). FR: reservoir frequency-dependent, FI: infectious frequency-dependent, MA: mass action (*: mass action with transmission rate dependent on pathogen load; immuno-epidemiological model), NA: not applicable (models with no explicit vector compartments); see section on Force of infection and Box 1 for details. ABM: agent-based model. All ABM models used individual animals as agents, except for Python Ndekou Tandong et al. (2020) [89] (**: agent-based modeling for animal mobility, with cities and trucks as agents exchanging animals, compartment model for transmission within cities.)

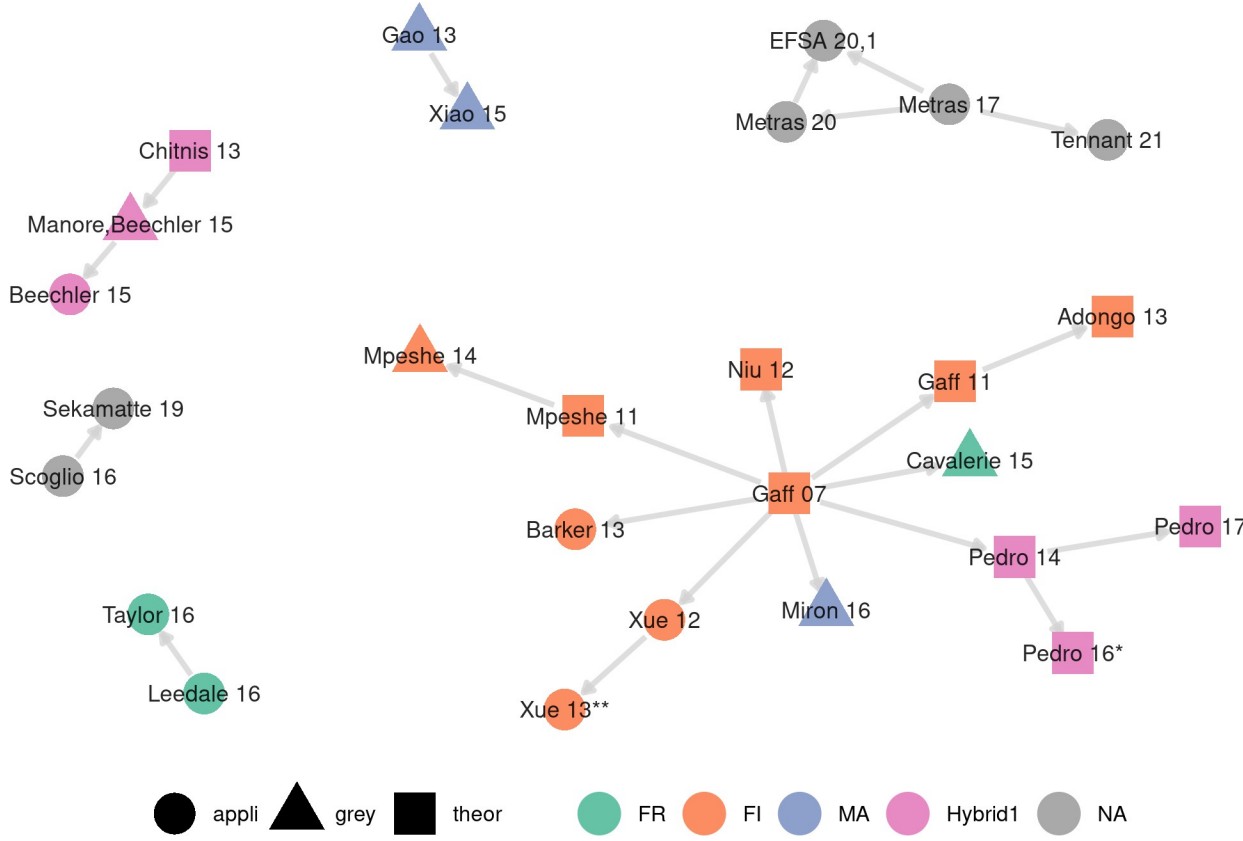

**Fig 2. Inspirational network of models.** Nodes are labeled with the reference of the associated studies (year abbreviated), shaped by model category, and colored by the functional form of the force of infection (FR: reservoir frequency-dependent, FI: infectious frequency-dependent, MA: mass action, NA: not applicable (models with no explicit vector compartments); see section on Force of infection and Box 1 for details). An edge between two nodes represents a model declaring the other as its parent model, as defined in the main text. Twenty-two models are not shown in this plot as they did not declare a parent model within the list of presently reviewed studies.*[75]. ** [76].

Métras et al. (2017) [78] and ran the parameter estimation algorithm on a new outbreak dataset. One of the two models described in EFSA AHAW Panel et al. (2020)[79] then made a stochastic model based on Métras et al. (2017, 2020) [77,78]. Tennant et al. (2021) [80] transformed the single-patch model of Métras et al. (2017) in Mayotte into a metapopulation model for the Comoros archipelago. Taylor et al. (2016) [81] used the model by Leedale et al. (2016) [82] set in Kenya and Tanzania to explore a new research question, i.e., to anticipate the effect of climate change in East African Community. Xiao et al. (2015) [83] modified the model by Gao et al. (2013) [84] to include seasonality through time-varying parameters.

## Contribution to the understanding of RVF epidemiology

**Objective of the modeling study.** To broadly describe the contribution of models to the study of RVF epidemiology, three main scientific objectives were identified (Table 1): exploring epidemiological mechanisms ('understand', n = 38), examining consequences of hypothetical outbreaks ('anticipate', n = 4), and assessing control strategies ('control', n = 7). In the present section, we focus on key features identified per objective.

The most common primary scientific objective of models was to understand epidemiological processes, in all model categories (from 72% of applied models to 79% of grey models, Table 1, Fig 3). Although in 11 cases, those models also aimed to anticipate or control

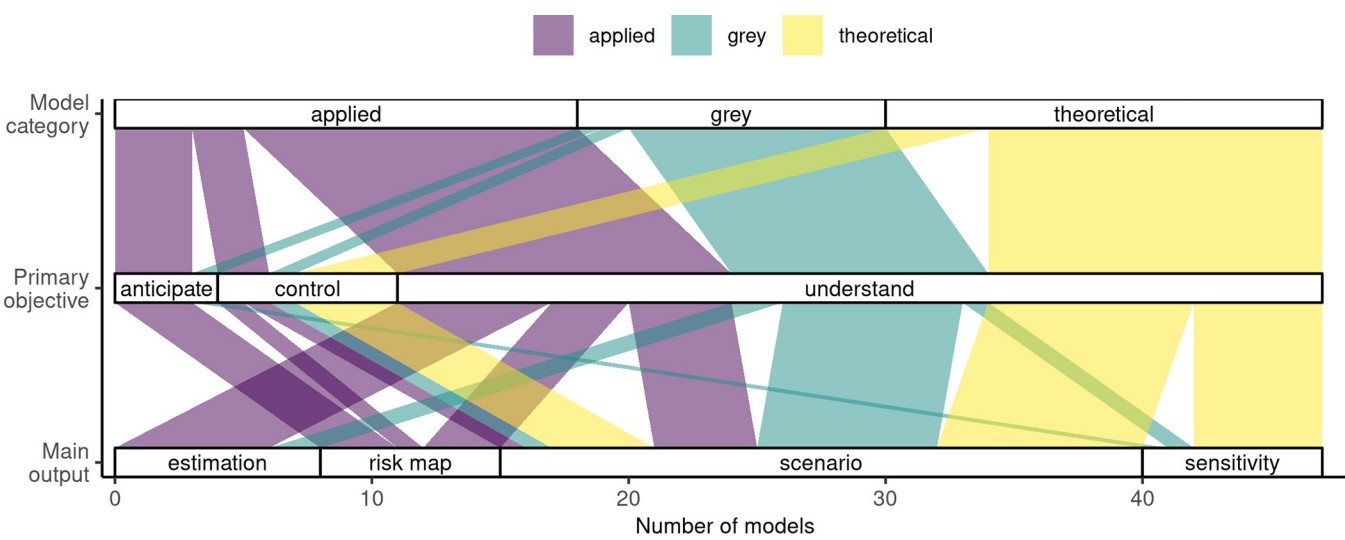

**Fig 3. Association between the model category, the primary objective of the study, and the main type of output chosen to illustrate the results.** This figure excludes two models for which the main output consisted of a deep analysis of the mathematical properties of the system (Table 1).

outbreaks in a secondary part [62,77,78,80,86–89,95,98,100]. In addition, in 30% of cases, model development in itself seemed to be a leading objective of the study. In such cases, contributing to RVF epidemiology was as important as contributing methodologically to RVF mechanistic modeling, by including for the first time a given compartment, parameter, or by developing a method to integrate data.

An interesting trend is the evolution of the objectives of modeling papers over the years, which increasingly include the control and anticipation of RVF outbreaks (15/26 studies in 2016-present, 7/23 in 2004–2015). Research on RVF, through mathematical modeling and other methods, has deeply enhanced our understanding of underlying epidemiological mechanisms, which now allows models to focus more on operational aspects. However, some papers did not formulate a precise research question and consequently did not tailor their model to a specific set of hypotheses or scenarios to test. Theoretical models have helped to broadly explore the pathosystem behavior when dealing with a lot of uncertainty, but such papers often lack clarity. A difficulty for theoretical papers is to convey how mathematical analysis can be helpful to field practitioners down the line [110]. Regarding applied and grey models, their specificity often relied on the geographical application and the dataset they used, rather than on a focused research question.

**Main outputs.** The main output of a model, holding the key message of the studies, could pertain to one of four main categories (Table 1): i) parameter estimation (n = 8), ii) risk maps (n = 7), iii) comparison of scenarios, defined as a small set of simulations with specific parameters varied (or processes turned off) across a small set of values (n = 25), and iv) sensitivity analysis, where a large subset (if not all) of parameters are varied across a large set of values (e.g., using sampling design to generate them), usually to produce an index quantifying the impact of each parameter on selected model outputs (n = 7). In two additional cases, the main results relied on a deep analysis of the mathematical properties of the system (e.g., van Kampen system-size expansion [98], Lyapunov exponent, Poincaré map [107]). A given paper could have produced several of these outputs but we tried to identify, with an inevitable part of subjectivity, the one standing out as the main output.

The main model output varied according to the model category and their primary objective (Table 1, Fig 3). Scenario comparison was the only main output used by all model categories

(Fig 3). Indeed, this type of analysis is flexible and can focus on a specific hypothesis and its impact on the system's behavior. Risk maps were only produced as a main output by applied models, and was the most common output for models with the aim to anticipate (Fig 3). Sensitivity analyses were mostly used by theoretical models as a main output (5/7), and never as such by applied models (Fig 3, 3/10 by grey models). Parameter estimation was mostly performed by applied models (6/8), and not at all by theoretical models (Fig 3, 2/8 by grey models). By nature, sensitivity analyses and parameter estimation are primarily done to understand the system better. Here, we highlight that theoretical and applied models can use different tools to contribute to a common objective. In three cases, parameter estimations were used further in the same model to help anticipate [78] or control [77,80] outbreaks, as a secondary objective. Fifty-five percent of models provided an estimation of a type of reproduction number, e.g., the basic reproduction number $R_0$, the effective reproduction number $R_e$, the seasonal reproduction number $R_{st}$ (phenomenological relationship estimated between environmental parameters and transmission rate), or the Floquet ratio $R_T$ (the expected number of cases caused by a primary case after one complete cycle of seasons [111]). Most of these reproduction numbers were obtained analytically (25/27). These estimates were highly variable and are therefore not reported here.

**Key questions.** Mechanistic models can help gauge the importance of hardly observable epidemiological processes, such as vertical transmission in vectors. This transmission route was included in around 50% of models, all having 'understand' as a main objective of the study. This seems representative of current knowledge on the importance of this process in the field. Indeed, evidence is limited regarding its potential role in the interepidemic maintenance of the virus [112]. Five models centered their research question on the quantification of this mechanism, in all categories (2 theoretical, 2 grey, 1 applied). Chitnis et al. (2013) [69] (theoretical) showed that while the vertical transmission rate does not impact $R_0$, it can contribute significantly to inter-epidemic persistence. Pedro et al. (2016) [75] (theoretical) estimated a linear and significant effect of vertical transmission on $R_0$ and vector eradication effort, although this effect became substantial only when vertical transmission rate was above 20% (percentage of infected mosquitoes' progeny which are infected). Such a rate seems much higher than what has been observed experimentally [113,114]. Manore & Beechler (2015) [66] (grey) focused on inter-epidemic activity in Kruger National Park (South Africa) and estimated that realistic vertical transmission rates should be combined with the presence of alternate hosts to allow RVF persistence. Lo Iacono et al. (2018) [93] (grey) showed that vertical transmission of RVFV in *Aedes* spp. was not a prerequisite for RVF persistence over time in Kenya. Durand et al. (2020) [91] (applied) concluded that vertical transmission could not be ruled out but nomadic herd movements were sufficient to explain the enzootic circulation of RVFV in Senegal. The inconsistent conclusions from those models might indicate a spatially and temporally heterogeneous role of vertical transmission in RVFV maintenance. Moreover, most models have considered a uniform vertical transmission rate. However, it is more likely that the percentage of infected progeny may vary depending on individuals. For instance, it has been evidenced in *Aedes dorsalis* the possible existence of 'stabilized infections' for the California encephalitis virus [115], i.e., a very small percent of mosquitoes are able to infect virtually 100% of their progeny, so that infection in mosquitoes is able to persist over several generations. Models could be used to explore this scenario, as the same mechanism has been suggested for RVFV [113].

The importance of animal movements in RVFV spread and persistence is another key question explored by included studies. Theoretical models show that local and distant spread of the virus are positively correlated to animal movement speed and flow size [83,89], but complex relationships exist in case of heterogeneous movements and livestock death rates across the network [105]. Spatial spread can also be limited by physical barriers to livestock migration

[65]. The role of animal movements in RVFV spread is highlighted by applied models, especially with a low transmission probability [87] or in a low vectorial capacity [86] context. Métras et al. (2017) [78] suggested that import of infected livestock in 2007 was a major driver of RVF emergence in Mayotte in 2008–2010, and Gao et al. (2013) [84] that a transport of only a few infectious animals from Sudan to Egypt could be sufficient to start an outbreak. Across the Comoros archipelago, RVFV seems to be able to persist even in the absence of new introductions, with Grande Comore and Moheli more likely to sustain local transmission without new viral introductions [80].

Original research questions stood out from the rest. Beechler et al. (2015) [67] studied the impact of co-infections with the mycobacterium causing bovine tuberculosis (BTB). Their data highlighted that RVFV infection was twice as likely in BTB+ than BTB- individuals. Once this effect was incorporated in a model, an increase in BTB prevalence nonlinearly affected three RVF outbreak metrics: the outbreak size in both BTB-infected and BTB-free populations, the timing of the peak, and the outbreak duration. Pedro et al. (2017) [106] looked at the possible role of ticks as vectors in addition to mosquitoes. They concluded that if ticks were capable of carrying and transmitting RVFV, this would sensibly change the transmission dynamics. Specifically, the size of outbreaks was increased, with a higher peak, reached faster, and the outbreak duration was reduced, compared to a situation with only mosquito vectors. It should be noted, however, that there is currently no evidence of the ability of ticks to biologically transmit the virus [116]. By contrast, other species which have been experimentally demonstrated as competent, either as biological (such as sandflies [117,118]), or mechanical vector [119] have not been included so far in RVF models. Tuncer et al. (2016) [97] developed an immuno-epidemiological model in which pathogen load impacted transmission rate, and focused on the identifiability of parameters (i.e., the uniqueness of parameter values able to reproduce a given model trajectory) rather than the epidemiological impact of such a hypothesis.

A single model [81] has looked at the possible effect of climate change on RVF risk, in Eastern Africa. This likely does not reflect a lack of interest for this issue, but could rather indicate that mechanistic modeling is not the preferred method to study such trends, compared to phenomenological (i.e., statistical) models [120–123]. In their review, Métras et al. (2011) [36] had highlighted the widespread use of phenomenological models to assess RVF risk across spatio-temporal scales. Phenomenological models can play a key role in selecting relevant processes to include or characterize suitable habitats, by highlighting significant correlations in complex datasets [124–126]. Such phenomenological models can then be nested into mechanistic models for specific processes (e.g., temperature-dependency, density-dependency). Mechanistic and phenomenological approaches can be seen as complementary ways to build a comprehensive view of vector-borne and zoonotic pathosystems [127]. Still, how to prioritize research on livestock and human health in the context of climate change is up to debate [128,129].

**Control measures.** Currently, vaccination against RVFV is only available for livestock, using live attenuated virus or inactivated virus vaccines, with limitations in their use [130]. Ten models reflected on possible vaccination strategies (Table A in S1 Text), in all categories (3 applied, 5 theoretical, 2 grey). The main objectives of all of these studies were to 'control', except for Métras et al. (2020) [77] for which it was a secondary objective. Such strategies were shaped by parameters such as the time to build-up immunity, vaccine efficacy, coverage, and regimen (Table A in S1 Text). Most models confirmed quantitatively the intuitive need for vaccination to happen before outbreaks or quickly after the first cases are detected, to have a significant impact (Table A in S1 Text). EFSA AHAW Panel et al. (2020—Model 1), Gachohi et al. (2016) and Métras et al. (2020) [77,79—Model 1,99] incorporated constraints on the

number of individuals vaccinated per day, so that a given coverage is reached at a realistic pace. Regarding the choice of hosts to vaccinate, Gachohi et al. (2016) [99] highlighted that while small ruminants needed a smaller coverage than cattle to achieve a given reduction in incidence, the vaccination of cattle provided the benefit of protecting both ruminant populations. This important role of cattle in RVFV transmission was due to a higher vector-to-host ratio and a larger body surface area, attracting more mosquitoes. Métras et al. (2020) [77] was the only model evaluating a possible human vaccination campaign. They estimated that, in the context of Mayotte island, vaccination of livestock was the most efficient strategy to limit human cases, compared to human vaccination. It required fewer doses than human vaccination to achieve a similar reduction in cases, assuming a highly immunogenic, single dose, and safe vaccine were available in both populations. This model took into account human exposure to livestock in their risk of infection. Adongo et al. (2013) [64] showed that optimal strategies differed depending on whether one prioritized the minimization of costs (doses) or of infections, with no clear take-home message for policy makers. Chamchod et al. (2016) [108] explored differences between the use of live and killed vaccines, and showed that due to the associated reversion of virulence, the use of live vaccines could render RVFV enzootic in situations where $R_0$ is initially below one.

Vector control methods, using adulticides or larvicides, are expensive and difficult to implement, due to the diversity of potential vector species and of larval developmental sites to treat [17,20]. These mitigations methods have been tested in a few models, with ambiguous results. Miron et al. (2016) [70] concluded that reducing mosquito lifetime under 8.7 days would reduce $R_0$ below one. In one study [63], both adulticides and larvicides were efficient to reduce the number of cases, when compared to no-intervention in a context of high virus transmission. In Mayotte, mosquito abundance had to be decreased by more than 40% to reduce RVF incidence and epidemic length, and an increased duration of epidemics was observed with lower levels of control [79—Model 1]. In the same model, vector control showed efficiency when coupled with culling strategy.

Few models considered movement restriction as a control method. A reduction of movements led to a decrease in disease spatial spread [86] and in incidence [95], and can help to eradicate the disease [89]. In Uganda, Sekamatte et al. (2019) [87] concluded that during periods of low mosquito abundance, movement restrictions led to a significant reduction in incidence. Movement restrictions had little impact in case of high vector abundance if used alone, and should therefore be combined with mosquito control. However, in some cases, mitigating measures could have unexpected consequences. In Comoros, scenarios of movement restriction between Grande Comore and other islands of the archipelago delayed the outbreak to a more suitable season, making it more severe overall [80]. By contrast, within-island control appeared to be more effective.

Testing and culling infected animals has been compared to other mitigations methods by three studies. This appeared to be one of the best strategies when conducted during 28 days after the detection of an outbreak in the theoretical model by Gaff et al. (2011) [63]. In the Netherlands, a RVFV-free area, a model concluded that stamping out in a 20 km radius around an outbreak could be the most effective strategy when comparing with scenarios of vaccination or other culling strategies [79—Model 2]. Nevertheless, in Mayotte, an effective strategy seemed hard to implement due to the high levels of animal testing and culling required [79—Model 1].

Overall, modeling studies often (6 applied, 6 theoretical, 3 grey) incorporate control-like scenarios, but the applicability of such simulations can be improved. Few models tried to assess RVF mitigation strategies in real endemic settings. Indeed, among six studies set in areas with history of RVFV circulation, only two had 'prevent' as a primary objective. Vaccination

(n = 10) and vector control (n = 5) were the main strategies considered by models, although they currently present major on-field limitations [20]. In addition, simulated vector control scenarios are often simplistic, consisting of a variation of one parameter homogeneously, and only one model distinguished the use of larvicides and of adulticides [63]. Finally, only five models considered movement restrictions as a mitigation strategy, which has been highlighted as a key determinant of RVFV spread and persistence in some epidemiological contexts [91]. Future efforts should focus on incorporating field constraints into their scenarios, while keeping in mind the transboundary nature of RVFV transmission [131–133].

## Model features

**Geographical context.** Locations of applied and grey models are mapped in Fig 4A. The scale of applied and grey models varied from local to international (Fig 4B). The sub-national scale was the most prevalent in both applied (10/18) and grey models (4/13) (Fig 4B). Regarding zones with known presence of RVF, several countries reporting numerous outbreaks in the last 15 years [20] have had at least one specific model developed (Burundi, Comoros, Kenya, Madagascar, Rwanda, Senegal, South Africa, Sudan, Tanzania, and Uganda). Besides, the Netherlands and the USA, both RVF-free, were also used as case studies for several models.

Spatial models, with at least two distinct locations, represented 45% (n = 22) of models (Table B in S1 Text). Among those, twelve were applied, five theoretical, and five grey models. All were discrete spatial models. Sixteen out of twenty-two (73%) spatial models incorporated connections between their spatial entities (Table B in S1 Text): vertebrate hosts moved in nine cases, vectors and hosts could move in three cases, and in four other cases, the connection was indirect, in the sense that the force of infection of one location was influenced by neighbors, taking into account distance, or prevalence. Three models were not spatialized but did include emigration and immigration of hosts (Table B in S1 Text).

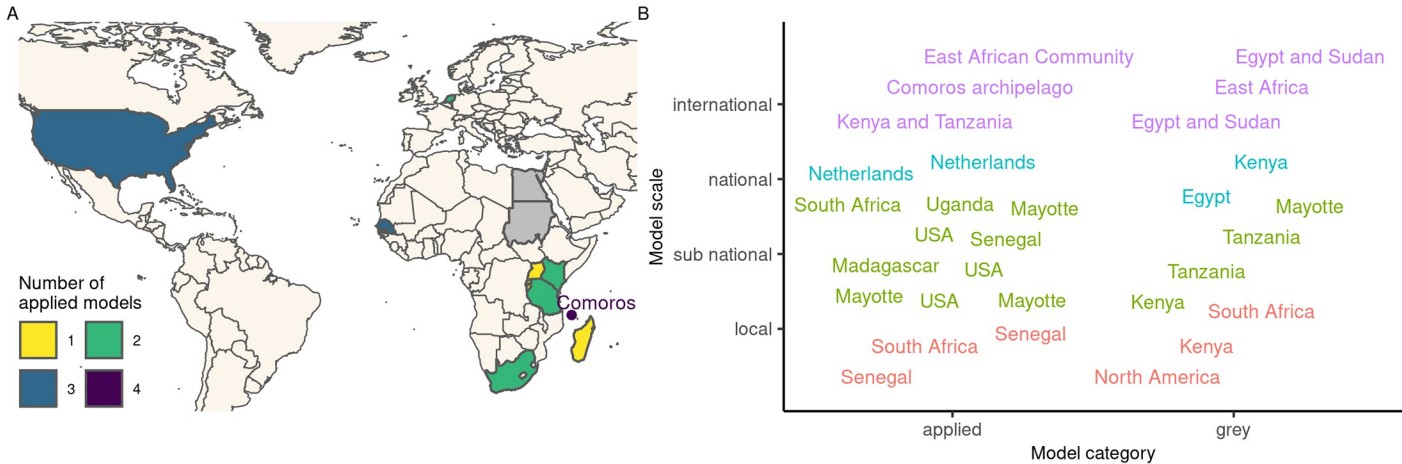

**Fig 4. A**—Geographical context and number of RVF models. Grey models are mapped but not counted in totals because they sometimes refer to a non-precise context (e.g., East Africa, North America, see B). Locations of grey models which are not also studied in applied models are shown in grey (Egypt, Sudan). The point north of Madagascar, accompanied by text, is centered on the Comoros archipelago. It stands for four models applied to Mayotte island and one model applied to the whole Comoros archipelago, including Mayotte. Map source: Natural Earth (https://www.naturalearthdata.com/). **B**—Scale of applied and grey models. Labels represent model locations, with one label per model, hence sometimes repeated locations. Labels are colored to help identify the scale (y-axis). East African Community = Burundi, Kenya, Rwanda, and Tanzania. Besides, all four models applied to Mayotte considered the whole island (374 sq. km), but those models are classified as sub-national. Sub-figures A and B are not restricted to spatial models (for those specifically, see Table B in S1 Text).

It should be noted that regions with recurring virus circulation, such as Botswana, Mauritania, Mozambique, or Namibia [20] are still left out from the RVF modeling effort. Identifying the possible hurdles preventing model development in those regions is important. In addition, RVF being a transboundary animal disease, larger scale models are needed, able to gauge the role of animal movements in the transmission dynamics. Currently, international applied models do not incorporate connections between their spatial entities (Table B in S1 Text), probably due to a lack of data. A coordinated data collection effort is required across affected countries, focusing on both commercial and pastoral mobility, and making these data easily accessible to epidemiological research teams.

**Data.**   Data were used in 25 out of 49 (51%) models. Here, we define data as any raw information, as opposed to a readily available parameter value extracted from another study. Several types of data were used (Table C in S1 Text): experimental (4/25), environmental (19/25), epidemiological (15/25), demographic (18/25), related to movements (6/25, Table B in S1 Text), and geographical (6/25). Most (23/25) models used more than one type of data, and sometimes had several distinct datasets per type (Table C in S1 Text). Among grey models, seven used data and six did not.

We identified a total of 102 datasets (Table C in S1 Text), corresponding to four datasets per model on average (102/25), ranging from two to ten. Only 44% of all datasets used by models incorporated a spatial dimension (measures in at least two distinct locations), and 45% a time dimension (measures for at least two different time points) (Table C in S1 Text). Regarding epidemiological datasets (n = 24), 25% were spatialized, and 58% were time-series (Table C in S1 Text). This is lower than environmental datasets (n = 28), which were 57% spatialized and 86% time-series (Table C in S1 Text). This supports conclusions made in recent reviews [24,25] which highlighted important gaps in RVF epidemiological data. Specifically, such gaps included the lack of fine-scale geographical metadata, preventing the study of within-country variation; the need for long-term studies in both endemic and non-endemic countries, to evaluate a possible increase in RVFV activity and exposure; and studies considering wildlife, livestock, and human concurrently, using standardized reporting and uniform case definitions [24,25]. Potential corrective measures would depend on whether such missing data are not collected or not made accessible. Most models with data managed to use at least one spatialized dataset (15/25, 60%) or time-series (23/25, 92%, Table C in S1 Text). This indicates that mechanistic models can resort to all types of data to try and compensate for the lack of precision in epidemiological reporting. Five studies used epidemiological data not published elsewhere [67,71,77,78,91], showing that modeling studies can also be seen as a way to valorize new datasets.

We categorized data use into three categories: calibration, input, and model assessment. Calibration was defined as the parametrization of one process or initial condition of the model, transforming the data in some way. This was done in 17 cases (Table C in S1 Text). Input was the fact of using the raw data directly as a parameter or initial condition of the model. This was done in 20 cases (Table C in S1 Text). Model assessment referred either to parameter inference or qualitative estimation looking to maximize similarity between epidemiological model outputs and data. This was done in 12 cases (Table C in S1 Text).

Ultimately, building accurate models helpful for policy makers requires the support of data. However, for RVF as well as for other infectious diseases, no single data source can be expected to inform each relevant parameter. Hence, the integration of information from many heterogeneous sources of data has become the norm [134]. This is a challenging task, as different datasets will be of different quality, potentially dependent, or in conflict [134]. Model-driven data collection can be a solution, but remains the exception rather than the rule [135]. Finally, we noted that in 40% of cases (4/10), models tailored to a location with known RVFV

circulation, and which used epidemiological data, did not include any scientist from a local institution in their author list. This is important to develop more realistic and useful models, and at a time when concerns are being raised about the equity of South-North research collaborations [26,136,137].

**Host and vector compartments.** Most models (34/49) included a single vertebrate host category, most of the time broadly labeled as livestock without making distinction between species (25/34, Table 1). When two hosts were accounted for, it was most often done to add a human compartment (9/14, Table 1). Cecilia et al. (2020), Durand et al. (2020), Fischer et al. (2013), and Gachohi et al. (2016)[88,91,92,99] distinguished small ruminants (sheep and goats) from cattle. This grouping was made to incorporate differences in attractiveness to mosquitoes [88,91,92,99] or in RVF-induced mortality [88,91,99]. In addition to livestock, Barker et al. (2013) [73] included birds as incompetent hosts, used as alternate blood-feeding sources by vectors, namely *Cx. tarsalis* and *Ae. melanimon*. The model by McMahon et al. (2014) [95] was the only one explicitly including a wildlife compartment, but did not describe the way the associated carrying capacity, (i.e., the maximum population size which can be sustained by the environment) was estimated based on land use data. Sumaye et al. (2019) [94] included a probability to pick up infection from wildlife hosts with a single parameter. Beechler et al. (2015) and Manore & Beechler (2015) [66,67] both modeled African buffaloes (*Syncerus caffer*), either captive or free-ranging.

The role of wildlife seemed largely understudied. Even if RVFV circulation has been highlighted in several wildlife species, with clinical signs in some ruminants, the potential role of those species in the epidemiological sylvatic cycles in endemic areas is still poorly understood [138–140]. Studying the competence of local wildlife species for RVFV transmission, along with their attractiveness to mosquitoes, is a prerequisite to determine the relevance of this question in a given territory [139,141–143].

In hosts, assumptions regarding the clinical expression of the disease varied. Chitnis et al. (2013), McMahon et al. (2014), Pedro et al. (2014), and Pedro et al. (2016) [69,75,95,107] included an asymptomatic state in hosts. Durand et al. (2020), Gachohi et al. (2016), Leedale et al. (2016), Taylor et al. (2016), and Tennant et al. (2021) [80–82,91,99] distributed hosts in age classes and (except Tennant et al. (2021) [80]) took into account differences in disease-induced mortality across classes. In Tennant et al. (2021) [80], only younger age classes moved between islands of the Comoros archipelago, and the initial proportion of immune individuals differed between classes. Cavalerie et al. (2015), Chamchod et al. (2014), Chamchod et al. (2016), Durand et al. (2020), and Sumaye et al. (2019) [71,91,94,100,108] incorporated abortion in livestock hosts due to RVFV infection.

In terms of transmission routes, Cavalerie et al. (2015), Durand et al. (2020) and Nicolas et al. (2014) [71,90,91] included the possibility of direct transmission between vertebrate hosts. Among eleven models including a human compartment (Table 1), nine considered livestock-to-human transmission by direct route (without vector) and ten models considered mosquito-to-human transmission. From these ten models, three [72,94,103] considered human-to-mosquito transmission. The low representation of this transmission route may reflect a confusion in the likely small role played by humans in the RVFV epidemiological cycle. As human-mosquito transmission has not been documented so far, humans may often be mistakenly considered as dead-end hosts [20,144]. Nevertheless, some data, while scarce, suggest they could develop a high viremia [144–147], which would be sufficient to infect mosquitoes. Under this hypothesis, humans could have a role in the long distance spread of the virus [148]. Considering this knowledge gap and the difficulty to obtain direct observations on that matter, it would seem relevant for future models to evaluate whether human-to-mosquito transmission is necessary to explain observed transmission dynamics.

Models with explicit vector compartments (43/49) included one (n = 20), two (n = 20), or more (n = 3) vector taxa (Table 1). Models with two taxa were all combining *Aedes* and *Culex* spp. vectors, while models with one vector taxon often did not specify the genus or species (10/20). The diversity of vectors was important among studies considering them at the species level, with the most often represented in models being *Ae. vexans* (n = 5), followed by *Cx. poicilipes* (n = 4). Pedro et al. (2017) [106] studied ticks (*Hyalomma truncatum*) in addition to *Aedes* and *Culex*. Sumaye et al. (2019) [94] included *Ae. mcintoshi*, *Ae. aegypti*, and two generic *Culex* vectors in their model, distributed in different ecological zones of Tanzania. Cecilia et al. (2020) [88] included *Ae. vexans*, *Cx. poicilipes*, and *Cx. tritaeniorhynchus* distributed in different ecological zones of Senegal.

Eleven models (22%) incorporated the influence of abiotic factors on the life cycle and competence of vectors, with dedicated equations. Cecilia et al. (2020), EFSA AHAW Panel et al. (2020 –Model 2), Fischer et al. (2013), Leedale et al. (2016), Lo Iacono et al. (2018), and Mpeshe et al. (2014) [72,79—Model 2,82,88,92,93] took into account the influence of temperature and/or rainfall on the lifespan of adult vectors. Gachohi et al. (2016), Leedale et al. (2016), Lo Iacono et al. (2018), Mpeshe et al. (2014), Xue et al. (2012), and Xue et al. (2013) [72,74,76,82,93,99] took into account the influence of temperature and/or rainfall on the egg laying rate, and on the development or survival of aquatic stages. Barker et al. (2013), Cecilia et al. (2020), Fischer et al. (2013), Lo Iacono et al. (2018) Mpeshe et al. (2014), and EFSA AHAW Panel (2020—Model 2) [72,73,88,92,93] took into account the influence of temperature on the extrinsic incubation period (EIP) and on the biting rate. Durand et al. (2020) [91] considered it on EIP only and Leedale et al. (2016) [82] on biting rate only. Fischer et al. (2013), Lo Iacono et al. (2018), Mpeshe et al. (2014), and Durand et al. (2020) [72,91–93] considered differences between *Culex* and *Aedes* mosquitoes for EIP and/or biting rate. Further sophistications, including the dependence to water body surface, were included into Cecilia et al. (2020), Durand et al. (2020), and in Lo Iacono et al. (2018) [88,91,93]. For Cecilia et al. (2020) and Durand et al. (2020) [88,91], this was done indirectly by relying on an external entomological model for vector population dynamics [149]. Overall, and due to the lack of data, modeling the impact of abiotic factors on the life cycle and competence of mosquitoes often relied on using data from different genera or species than those under study. In such cases, authors considered this choice preferable to a constant parameter or an arbitrary mathematical function.

Modelers are often faced with a substantial lack of data on vector presence and population dynamics when parameterizing their model. In Métras et al. (2017), Métras et al. (2020), and Tennant et al. (2021) [77,78,80], the lack of data on vector densities urged the authors to use an environmental proxy (Normalized Difference Vegetation Index (NDVI) or rainfall) to drive vectorial transmission, without including an explicit vector compartment. This type of data have been used previously to map RVFV transmission risk [150,151].

In reviewed models, the only source of variability in the feeding behavior of vectors was the inclusion of trophic preference for one host species over the others [88,91,92,99]. However, studies have suggested that the infected or uninfected status of the host might also play a role, for different pathogens [152,153], including for RVFV [154,155]. Future models could incorporate this mechanism to test its epidemiological importance.

Dealing with multiple hosts and vectors makes it difficult to predict disease emergence, spread, and potential for establishment. It has been shown that accounting for a higher biodiversity in epidemiological models can result in amplification or dilution effects depending on species' competence and abundance [156,157]. In the case of RVFV, the role and contribution of hosts and vectors to transmission dynamics is largely understudied. Quantifying these roles is crucial to design targeted and efficient control strategies, and will require more knowledge on the intrinsic heterogeneity between host and vector species. Within-host and within-vector

modeling can help in this matter, but such models for RVF are rare [97,158]. Besides, a paramount hypothesis driving model behavior is the contact structure assumed between hosts and vectors, mathematically embodied by the force of infection.

**Force of infection.** We chose to focus on the diversity of functional forms (FFs) used in RVF models for the force of infection related to vector-borne transmission. This was applied only to models explicitly including a vector compartment. Among those, a majority (29/43) did not justify their choice of FF, even though the force of infection, as a disease transmission term, encapsulates authors' assumption on the host-vector interactions, and therefore influences their predictions (Fig 5, [159]).

Six FFs were found in reviewed models (Table 1, Fig 5). We detail them in Box 1. Thirteen models used a reservoir frequency-dependent FF (Eq 1 in Box 1, Table 1, Fig 5). Eight models used a mass action FF (Eq 2 in Box 1, Table 1, Fig 5). Twelve models used an infectious frequency-dependent FF (Eq 3 in Box 1, Table 1, Fig 5). Ten models used alternative FFs, which all intended to avoid the shortcomings of other FFs by introducing parameters to constrain the contact rate between host and vector populations (Eqs 4–6 in Box. 1, called Hybrid1 (n = 8), Hybrid2 (n = 1), and Hybrid3 (n = 1) in Table 1 and Fig 5).

---

### Box 1: Diversity of assumptions and functional forms for the force of infection in models of RVFV transmission dynamics

In standard susceptible-infected-recovered (SIR)-type models, the force of infection (FOI) is the rate at which individuals go from the susceptible (S) state to the infectious (I, or exposed, E) state. Biologically, the FOI can be decomposed as $p_{contact} \cdot p_{inf} \cdot p_{transm}$. For vector-borne transmission, $p_{contact}$ is the contact rate between vectors (subscript $v$) and hosts (subscript $h$), $p_{inf}$ is the probability that a given contact is with an infectious individual, and $p_{transm}$ is the probability that a contact with an infectious individual results in successful transmission. This can be declined in two directions of transmission: vector-to-host and host-to-vector, which affects the value of these parameters. For $p_{inf}$, under the hypothesis of homogeneous mixing, we have:

$$p_{inf,v \to h} = \frac{I_v}{N_v}$$

$$p_{inf,h \to v} = \frac{I_h}{N_h}$$

The value of $p_{transm}$ can also vary depending on the source and target of the infection, but is not linked to host nor vector densities, but rather individual-level parameters (e.g., species, viremia, immune response). The different functional forms which can be seen in vector-borne disease models then arise from different assumptions on $p_{contact}$ [160].

### Reservoir frequency-dependence

The reservoir frequency-dependent (FR, n = 13, Eq 1) functional form assumes that the rate at which a vector bites hosts is constant across host (reservoir) densities (i.e., the vector does not feed more if there are more hosts), while the number of bites received by a host is proportional to the current vector-to-host ratio (i.e., a host is fed upon more if surrounded by more mosquitoes, at constant host population). Consequently, we get:

- $p_{contact,h \to v} = a$, with $a$ being the biting rate, usually defined as the maximal rate allowed by the gonotrophic cycle (i.e the minimum time required between blood meals for a female to produce and lay eggs). This results in $FR_{h \to v} = a \cdot \left(\frac{I_h}{N_h}\right) \cdot p_{transm,h \to v}$.

- $p_{contact,v \rightarrow h} = a. \frac{N_v}{N_h}$ which simplifies with $p_{inf,v \rightarrow h}$ and results in

$$FR_{v \rightarrow h} = a. \left( \frac{I_v}{N_h} \right) .p_{transm,v \rightarrow h}.$$

We can write, using $\beta_{hv}$ and $\beta_{vh}$ as aggregated terms, sometimes called adequate contact rates as in Gaff et al. (2007) [62]:

$$FR_{h \rightarrow v} = \beta_{hv}. \frac{I_h}{N_h}$$

$$FR_{v \rightarrow h} = \beta_{vh}. \frac{I_v}{N_h} \tag{1}$$

This functional form (FF) is therefore called reservoir frequency-dependent because the total number of hosts is on the denominator for both transmission directions ($v \rightarrow h$ and $h \rightarrow v$). With this FF, the vector-to-host transmission rate linearly increases with the vector-to-host ratio, and can therefore reach unrealistic values. Indeed, at some point, hosts are expected to deploy defense mechanisms to protect themselves from biting, preventing the vector population from getting all the blood meals needed.

## Mass action

The mass action functional form (MA, n = 8, Eq 2), sometimes called pseudo mass action, is density-dependent. It assumes that a vector bites hosts at a rate proportional to the number of hosts, and that a host is bitten at a rate proportional to the number of vectors. Consequently, we get:

- $p_{contact,h \rightarrow v} \propto N_h$, ignoring a possible constant derived from the previous biting rate $a$, which simplifies with $p_{inf,h \rightarrow v}$ and results in $MA_{h \rightarrow v} \propto I_h. p_{transm,h \rightarrow v}$.

- $p_{contact,v \rightarrow h} \propto N_v$, which similarly gives $MA_{v \rightarrow h} \propto I_v. p_{transm,v \rightarrow h}$.

$$MA_{h \rightarrow v} = \beta_{hv}.I_h$$

$$MA_{v \rightarrow h} = \beta_{vh}.I_v \tag{2}$$

With this functional form, the biting rate of vectors per unit time can exceed their physiological capacity above certain host densities, which again, becomes unrealistic.

## Infected frequency-dependence

Following the nomenclature by Wonham et al. (2006) [159], who presented susceptible frequency dependence, we describe the infectious frequency-dependent (FI, n = 12, Eq 3) functional form. It assumes that the rate at which a vector bites hosts is constant across host (reservoir) densities, while the number of bites received by a host is constant across vector densities. The transmission terms are then both correlated to the proportion of infectious in the population:

$$FI_{h \rightarrow v} = \beta_{hv}. \frac{I_h}{N_h}$$

$$FI_{v \to h} = \beta_{vh} \cdot \frac{I_v}{N_v} \tag{3}$$

The only plausible situation inducing a constant contact rate in both directions is one where the vector-to-host ratio remains constant. Indeed, if we assume that a vector systematically gets the blood meals it physiologically needs, and modeled hosts are the sole source of blood, then a shortage in hosts (high vector-to-host ratio) should result in an increase in bites per host (FR functional form). Alternatively, if the number of bites received per host is saturated (and constant) to account for their defense mechanism, then a vector's biting rate should vary with host densities, depending on whether this constrained system allows it to feed as it needs.

## Alternative functional forms

Functional forms FR, MA, and FI can only apply biologically at certain population densities, outside of which they can generate aberrant values and therefore lead to erroneous predictions [159]. Three alternative FFs were found in RVF models to prevent this issue (Fig 5). Those FFs require additional parameters to constrain the contact rate between populations.

The first alternative FF (Eq 4, Hybrid1 in Table 1 and Fig 5, n = 8) was first used in a RVF model by Chitnis et al. (2013) [69], who previously formulated it in a model of malaria transmission [161].

$$Hyb_{1,h \to v} = \frac{\sigma_v \sigma_h N_h}{\sigma_v N_v + \sigma_h N_h} \cdot \alpha_{vh} \frac{I_h}{N_h}$$

$$Hyb_{1,v \to h} = \frac{\sigma_v \sigma_h N_v}{\sigma_v N_v + \sigma_h N_h} \cdot \alpha_{hv} \frac{I_v}{N_v} \tag{4}$$

In Eq 4, $\alpha_{hv}$ and $\alpha_{vh}$ refer to probabilities of successful transmission given contact, from host to vector and vice versa. $\sigma_v$ is defined as the maximum number of times a mosquito would bite a host per unit time, if freely available. This is a function of the mosquito's gonotrophic cycle and its preference for a given host species. $\sigma_h$ is the parameter added to avoid abnormally high contact rates and represents the maximum number of bites sustained by a host per unit time. Although $\sigma_h$ seems virtually impossible to estimate in the field, this alternative FF can efficiently prevent erroneous model predictions and has therefore often been reused in RVF models. It is also the most justified FF (5/8, Fig 5). Some slight variations in its mathematical formulation can be found in Sumaye et al. (2019) [94].

The second alternative FF was used by McMahon et al. (2014) [95] (Eq 5, Hybrid2 in Table 1 and Fig 5, n = 1).

$$Hyb_{2,h \to v} = inf_h \cdot sus_v \cdot e^{-\frac{t}{a}} \cdot I_h$$

$$Hyb_{2,v \to h} = inf_v \cdot sus_h \cdot e^{-\frac{t}{a}} \cdot I_v$$

$$r = -\sqrt{\frac{N_v + N_h}{A}} \qquad (5)$$

Here, *inf* and *sus* refer to a vector or host infectivity and susceptibility, respectively. The contact rate is formulated as $e^{\frac{-r}{a}}$, with *a* the characteristic length of local spread. In *r*, *A* is the patch area.

A last alternative FF was used in Lo Iacono et al. (2018) [93] (Eq 6, Hybrid3 in Table 1 and Fig 5, n = 1).

$$Hyb_{3,h \to v} = \alpha_{hv} . \tilde{\theta} \frac{I_h}{N_h}$$

$$Hyb_{3,v \to h} = \alpha_{vh} . m . \tilde{\theta} \frac{I_v}{N_v}$$

$$\tilde{\theta} = \frac{\theta}{1 + \frac{m}{q}}$$

$$m = p_f . \frac{N_v}{N_h} \qquad (6)$$

Here, $p_f$ is the proportion of the mosquito population able to detect and feed on the host species under consideration, and *m* is therefore an 'effective' vector-to-host ratio. $\tilde{\theta}$ is the biting rate, function of *m*, as well as of the rate of completion of the gonotrophic cycle $\theta$, and of *q*, the vector-to-host ratio for which vector fecundity is divided by two. This is done to account for the decrease in fecundity in the case of absence of sufficient hosts to take a blood meal.

In 82% of cases (14/17), a model used the same FF for its force of infection as its parent model (Fig 2). In 9/14 cases, the parent model did not justify the choice of FF used, and no further justification was provided in the subsequent model in 7/9 cases. In 3/17 cases, the FF was changed compared to the parent model, which induced a justification in 2/3 cases. In addition, in three cases, the representation of vectors was implicit in a model and its parent model, therefore preventing the classification of the force of infection into any FF.

Several review papers on various epidemiological models concluded that the choice of a functional form for the force of infection could greatly affect model behavior. Begon et al. (2002), Hoch et al. (2018), and McCallum et al. (2001) [160,162,163] focused on non-vectorial transmission. Hopkins et al. (2020) [164] focused on parasite transmission, which could be through a vector, but did not include possible variations in frequency-dependent functions. Wonham et al. (2006) [159] focused on FFs used to model vectorial transmission of West Nile virus and also noticed an important diversity. In 2001, McCallum et al. were already recommending to "explicitly state and justify the form of transmission used" as well as "evaluate

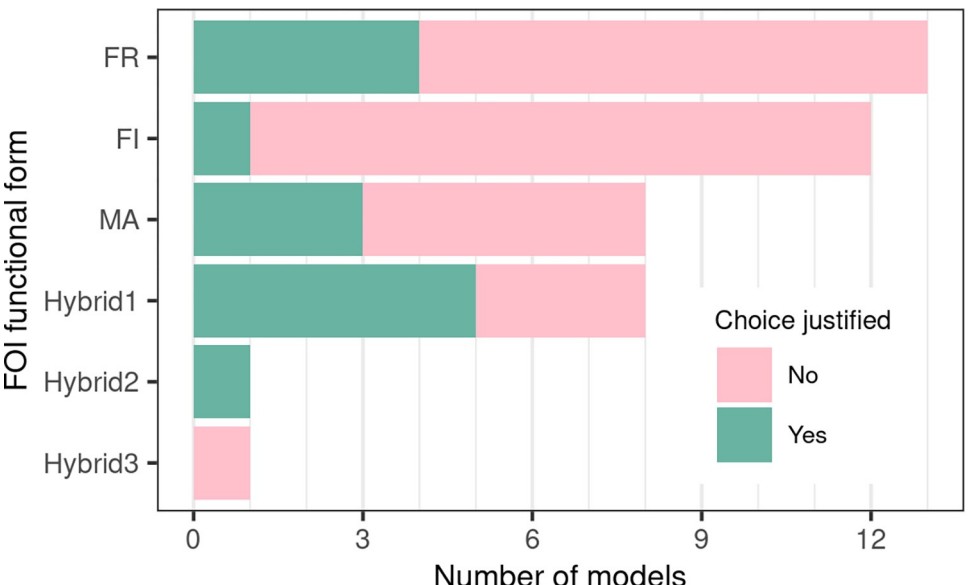

**Fig 5. Functional forms (FFs) used by models for their force of infection (FOI; vector-borne transmission only).**
FR: reservoir frequency-dependent, FI: infectious frequency-dependent, MA: mass action; see section on Force of infection and Box 1 for details. The full bar length indicates the number of models using a given FF, the color determines how many models properly justified their choice of FF. See Eqs 1–6 for details on each FF. See Table 1 for details on papers using a given FF.

several alternative models of transmission, if possible" [163]. Contact structures between host and vector populations are hard to observe in natural conditions. This should be an additional incentive for modelers to explicitly state the reasoning behind their choice of functional form, which can be motivated by the context of their case study (e.g., expected host and vector densities, mixing between the populations). A comparison of the FF listed presently would be useful. The conclusions might vary depending on whether this is done through theoretical scenarios, keeping all other parameters equal, or by fitting different models to a common empirical dataset. The latter might not be able to discriminate between FF to select the best-performing one, because of underlying correlations between parameters.

## Conclusion

In the last 5 years, more mechanistic models of RVFV transmission dynamics have been published (n = 26) than in the 10 previous years combined (n = 23). This possibly indicates a growing interest for RVF epidemiology, although it is known that the number of publications is continuously growing in all fields [26,165,166]. Our review highlighted important knowledge gaps, rarely addressed in mechanistic models of RVFV transmission dynamics. In our opinion, the most pressing issues are i) the incorporation of heterogeneity among host and vector species, in order to determine their relative role in transmission dynamics, which will require a focus at the within-host and within-vector scales, and ii) the development of large scale models, able to quantify the role of animal mobility in RVFV spread. Both of these research avenues will rely on novel data sets being generated, and will require methodological accuracy and transparency, particularly with regards to the choice of force of infection [113]. Indeed, as it reflects assumptions made on the contact rate between host and vector populations, this choice crucially influences model predictions and therefore cannot be made lightly. This systematic

review showed that, as was the case for West Nile virus [159], models of RVFV transmission dynamics make very distinct assumptions which render their results not directly comparable. We detailed them didactically, hoping to guide future models focusing on vector-borne transmission.

This increasing number of models could also reflect a growing trust in mechanistic models in the field of infectious disease epidemiology [167,168]. When it comes to decision-making for disease management, we agree with previous work showing that combining models is the most sensible approach rather than attempting to find the best model [169,170]. Indeed, the diversity of models' structure and hypotheses is a richness, which can be used to highlight actions that are robust to model uncertainty, but also identify key differences needing clarification through additional field exploration [169,170].

Importantly, we note that only seven studies made their code available (Table 1), which represents 23% of models published since 2015. Adopting this practice more broadly would increase the reproducibility of results and encourage the community to bring existing work further [110].

## Supporting information

**S1 Text. Reading grid and complementary tables.** Text A: Reading grid; Table A: Vaccination strategies implemented in models and main results; Table B: Characteristics of spatial models as well as non-spatial models with external renewal; Table C: Type of datasets and their use.
(DOCX)

Code used to produce figures and summary statistics is available in Github public repository at https://github.com/helenececilia/riftvalleyfever-model-review.git.

## Author Contributions

**Conceptualization:** Hélène Cecilia, Alex Drouin, Raphaëlle Métras, Benoit Durand, Véronique Chevalier, Pauline Ezanno.

**Data curation:** Hélène Cecilia, Alex Drouin.

**Formal analysis:** Hélène Cecilia, Alex Drouin.

**Investigation:** Hélène Cecilia, Alex Drouin.

**Methodology:** Hélène Cecilia, Alex Drouin, Raphaëlle Métras, Benoit Durand, Véronique Chevalier, Pauline Ezanno.

**Software:** Hélène Cecilia, Alex Drouin.

**Supervision:** Raphaëlle Métras, Thomas Balenghien, Benoit Durand, Véronique Chevalier, Pauline Ezanno.

**Validation:** Hélène Cecilia, Alex Drouin, Raphaëlle Métras, Thomas Balenghien, Benoit Durand, Véronique Chevalier, Pauline Ezanno.

**Visualization:** Hélène Cecilia, Alex Drouin.

**Writing – original draft:** Hélène Cecilia, Alex Drouin.

**Writing – review & editing:** Hélène Cecilia, Alex Drouin, Raphaëlle Métras, Thomas Balenghien, Benoit Durand, Véronique Chevalier, Pauline Ezanno.

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
