## [Decision Letter · Decision Letter 0]

19 May 2022

Dear Mr. Drouin,

Thank you very much for submitting your manuscript "Mechanistic models of Rift Valley fever virus transmission dynamics: A systematic review" for consideration at PLOS Neglected Tropical Diseases. As with all papers reviewed by the journal, your manuscript was reviewed by members of the editorial board and by several independent reviewers. In light of the reviews (below this email), we would like to invite the resubmission of a significantly-revised version that takes into account my and the reviewers' comments. 

General comment: I found this to be an interesting review of the various models of RVF, but as I am not a modeler, I was a little concerned as I could not tell which of the models cited were good and which were not, i.e., simply stating that “Eleven models (22%) incorporated the influence…” (line 496) doesn’t indicate how well these models did. One of the models incorporated ticks (see my comment 5 below) and I am not certain about many of the assumptions made on potential vectors. I also made a few minor editorial comments.

Specific comments:

1. Line 44: Yes, RVFV is primarily associated with Aedes and Culex mosquitoes, but numerous species of other mosquito genera have been found to be naturally infected in the field and shown to be competent vectors of RVFV in the laboratory. Depending on the situation, these species may play a critical role.

2. Line 58: All of the papers cited deal with European mosquitoes, but the sentence also says, “North America.” Why not also include “Turell MJ, Dohm DJ, Mores CN, Terracina L, Wallette DL Jr, Hribar LJ, et al. Potential for North American mosquitoes (Diptera: Culicidae) to transmit Rift Valley fever virus. J Am Mosq Control Assoc. 2008;24:502-7?”

3. Line 276: What do you mean by a vertical transmission rate of 10%? I assume that you mean than 10% of the orally infected mosquitoes produced at least 1 infected progeny and not that 10% of the progeny of the infected mosquitoes were infected. Given that vertical transmission is very rare to the first ovarian cycle progeny, and most of the infected mosquitoes won’t survive to produce a second cycle, how are you calculating this? What about stabilized infections, i.e., where in a very small percent of the mosquitoes, virtually 100% of their progeny, including first ovarian cycle progeny, are infected, and this persists for many, many generations. This is likely what allows RVFV to remain enzootic in a particular region.

4. Line 277: Why now use “transovarial” instead of vertical transmission. Yes, transovarial is likely to be the most important route of vertical transmission, but there is also transovum transmission. I would only use “vertical transmission.”

5. Line 295: Yes, Pedro et al. (2017) looked at what might happen if ticks could transmit RVFV. However, the last sentence of their abstract is, “These findings suggest that if ticks are capable of transmitting the virus, they may be contributing to disease outbreaks and endemicity.” Has anyone ever shown that an orally exposed tick, of any species, is able to transmit RVFV? If not, then why not include mites, lice, and various biting flies?

6. Line 336-338: Here the authors are stating that if “even with R0 < 1, RVF was likely to become endemic when live vaccines were implemented.” Do they mean that if a killed vaccine was used that RVF would NOT become enzootic? Note, as RVF is primarily a disease of animals, then shouldn’t it be enzootic, rather than endemic?

7. Line 340: Yes, everyone refers to larval developmental sites as “breeding” sites, but “breeding” has a sexual connotation, and mosquitoes do not “breed” in these locations. It would be more accurate to refer to them as “larval developmental sites.”

8. Line 355-357: Was it the “movement restriction between Grande Comore and other islands of the archipelago led to a more severe and delayed outbreak” or were other factors also involved that actually caused the more severe outbreak?

9. Lines 453-454: Yes, “livestock” does appear to be a single vertebrate host, but what if the area had cattle, goats, and sheep? These would all be considered livestock, but a certainly not a single vertebrate host.

10. Lines 458-459: Yes, as incompetent vertebrate hosts, birds might interfere with RVFV transmission. However, many of the mosquitoes, particularly the Aedes, that are involved in RVFV transmission preferentially feed on mammals. Therefore, whether birds were present or not would make little difference. This is unlike the situation with cattle as an alternative host for Anopheles mosquitoes and malaria, where the presence of cattle near a home may provide an attractive alternative host that was incompetent for human malaria and thus reduces malaria in the area.

11. Line 486: Why would humans be a dead-end host? Viremias in humans are extremely high, often more than 8 logs, i.e., higher than often observed in adult cattle, goats, or sheep. This is unlike viruses such as West Nile virus and Japanese encephalitis virus in which humans produce such a low viremia that they are indeed dean-end hosts.

12. Line 491: Why would Ae. vexans be the most common in these models? The very few studies on the ability of Ae. vexans orally exposed to RVFV to transmit RVFV indicate that there is a very wide range in vector competence for this species, ranging from <1% in the northern U.S., 5% in Europe, to about 20% in the southeastern U.S. I know of no studies that have determined if African strains of Ae. vexans are competent vectors. Note, isolation of virus from a field-collected mosquito does NOT mean that it is capable of transmitting that virus. Various arboviruses, including RVFV, have been detected in mosquito species that have been shown to be incompetent and thus not vectors.

13. Lines 504-506: yes, these models may have taken this into account, but the effect differs for different vectors, so how did they do that?

14. Lines 539-589: This is not as simple as it appears. For example, if 1 in 50 of the sheep in a flock is infected, most models would assume that 1 in 50 mosquitoes feeding on an animal in that flock would be exposed to virus. That is not correct. RVFV-infected animals are more attractive to a blood-seeking mosquito than their uninfected siblings, so greater than 1 in 50 would have been exposed to virus. Similarly, many mosquitoes may be dislodged before they obtain a blood meal, but mosquitoes are able to obtain a blood meal more rapidly from a RVFV-infected animal than from an uninfected member of the same species. Again, the successful feeding rate on RVFV-infected animals would be higher than the percent of animals actually infected. Have either of these been incorporated in the models?

15. References: Please ensure that all references are formatted correctly.

 a. In manuscript titles, only the first word and all proper nouns should be capitalized. See references 1, 14 and others where “rift valley fever.” should have been “Rift Valley fever.” See also references 7, 8, 10, and others where each word is capitalized, even if not a proper noun. Please check all of the references.

 b. See references 41 and 46. Is it PLOS ONE or PLoS ONE?

 c. What does the “Larchmt N” mean in reference 54?

 d. Why do you list the editor for some of the PLoS journals?

 e. Please go through all of the references and ensure that they are properly formatted.

Minor comments:

16. Line 44: Both “Aedes” and “Culex” should be in italics.

17. Line 91: RVFV was established non line 57 and should be used here. See also lines 278, 290, 376, 466 etc.

18. Line 151, Figure 1: The first block indicates that 372 records were identified (236 + 136). The next block indicates that 123 of these were removed prior to screening. So why weren’t 249, instead of 248 screened? In all of the other reductions, the numbers do match?

19. Line 169: Shouldn’t “4 cases…” be “four cases…” See also lines 367, 391-394, 421, etc. I am fine with using a numeral within a parentheses, but you need to be consistent outside of a parentheses.

20. Line 342: Is it really necessary, or even accurate, to present lifetime to a hundredth of a day? Why not simply state, “under 8.7 days…”

21. Line 448: Why is 4/10 39% instead of 40%?

22. Line 480: Is it RVF infection or RVFV infection? This occurs throughout the manuscript. Mosquitoes do not transmit RVF, they transmit the virus that causes RVF.

23. Lines 499-500: Shouldn’t “on the lifespan of adult vectors lifespan” be “on the lifespan of adult vectors?”

24. Line 530: What is a SIR-type model? I assume that the authors are referring to a susceptible-infected-recovered model, but don’t know if the reader will get that. Also, in Table 1, is it “sensible” or “susceptible?” I think that susceptible is more accurate.

We cannot make any decision about publication until we have seen the revised manuscript and your response to the reviewers' comments. Your revised manuscript is also likely to be sent to reviewers for further evaluation.

Sincerely,

Michael J Turell, Ph.D.

Associate Editor

Robert Reiner

Deputy Editor

General comment: I found this to be an interesting review of the various models of RVF, but as I am not a modeler, I was a little concerned as I could not tell which of the models cited were good and which were not, i.e., simply stating that “Eleven models (22%) incorporated the influence…” (line 496) doesn’t indicate how well these models did. One of the models incorporated ticks (see my comment 5 below) and I am not certain about many of the assumptions made on potential vectors. I also made a few minor editorial comments.

Specific comments:

1. Line 44: Yes, RVFV is primarily associated with Aedes and Culex mosquitoes, but numerous species of other mosquito genera have been found to be naturally infected in the field and shown to be competent vectors of RVFV in the laboratory. Depending on the situation, these species may play a critical role.

2. Line 58: All of the papers cited deal with European mosquitoes, but the sentence also says, “North America.” Why not also include “Turell MJ, Dohm DJ, Mores CN, Terracina L, Wallette DL Jr, Hribar LJ, et al. Potential for North American mosquitoes (Diptera: Culicidae) to transmit Rift Valley fever virus. J Am Mosq Control Assoc. 2008;24:502-7?”

3. Line 276: What do you mean by a vertical transmission rate of 10%? I assume that you mean than 10% of the orally infected mosquitoes produced at least 1 infected progeny and not that 10% of the progeny of the infected mosquitoes were infected. Given that vertical transmission is very rare to the first ovarian cycle progeny, and most of the infected mosquitoes won’t survive to produce a second cycle, how are you calculating this? What about stabilized infections, i.e., where in a very small percent of the mosquitoes, virtually 100% of their progeny, including first ovarian cycle progeny, are infected, and this persists for many, many generations. This is likely what allows RVFV to remain enzootic in a particular region.

4. Line 277: Why now use “transovarial” instead of vertical transmission. Yes, transovarial is likely to be the most important route of vertical transmission, but there is also transovum transmission. I would only use “vertical transmission.”

5. Line 295: Yes, Pedro et al. (2017) looked at what might happen if ticks could transmit RVFV. However, the last sentence of their abstract is, “These findings suggest that if ticks are capable of transmitting the virus, they may be contributing to disease outbreaks and endemicity.” Has anyone ever shown that an orally exposed tick, of any species, is able to transmit RVFV? If not, then why not include mites, lice, and various biting flies?

6. Line 336-338: Here the authors are stating that if “even with R0 < 1, RVF was likely to become endemic when live vaccines were implemented.” Do they mean that if a killed vaccine was used that RVF would NOT become enzootic? Note, as RVF is primarily a disease of animals, then shouldn’t it be enzootic, rather than endemic?

7. Line 340: Yes, everyone refers to larval developmental sites as “breeding” sites, but “breeding” has a sexual connotation, and mosquitoes do not “breed” in these locations. It would be more accurate to refer to them as “larval developmental sites.”

8. Line 355-357: Was it the “movement restriction between Grande Comore and other islands of the archipelago led to a more severe and delayed outbreak” or were other factors also involved that actually caused the more severe outbreak?

9. Lines 453-454: Yes, “livestock” does appear to be a single vertebrate host, but what if the area had cattle, goats, and sheep? These would all be considered livestock, but a certainly not a single vertebrate host.

10. Lines 458-459: Yes, as incompetent vertebrate hosts, birds might interfere with RVFV transmission. However, many of the mosquitoes, particularly the Aedes, that are involved in RVFV transmission preferentially feed on mammals. Therefore, whether birds were present or not would make little difference. This is unlike the situation with cattle as an alternative host for Anopheles mosquitoes and malaria, where the presence of cattle near a home may provide an attractive alternative host that was incompetent for human malaria and thus reduces malaria in the area.

11. Line 486: Why would humans be a dead-end host? Viremias in humans are extremely high, often more than 8 logs, i.e., higher than often observed in adult cattle, goats, or sheep. This is unlike viruses such as West Nile virus and Japanese encephalitis virus in which humans produce such a low viremia that they are indeed dean-end hosts.

12. Line 491: Why would Ae. vexans be the most common in these models? The very few studies on the ability of Ae. vexans orally exposed to RVFV to transmit RVFV indicate that there is a very wide range in vector competence for this species, ranging from <1% in the northern U.S., 5% in Europe, to about 20% in the southeastern U.S. I know of no studies that have determined if African strains of Ae. vexans are competent vectors. Note, isolation of virus from a field-collected mosquito does NOT mean that it is capable of transmitting that virus. Various arboviruses, including RVFV, have been detected in mosquito species that have been shown to be incompetent and thus not vectors.

13. Lines 504-506: yes, these models may have taken this into account, but the effect differs for different vectors, so how did they do that?

14. Lines 539-589: This is not as simple as it appears. For example, if 1 in 50 of the sheep in a flock is infected, most models would assume that 1 in 50 mosquitoes feeding on an animal in that flock would be exposed to virus. That is not correct. RVFV-infected animals are more attractive to a blood-seeking mosquito than their uninfected siblings, so greater than 1 in 50 would have been exposed to virus. Similarly, many mosquitoes may be dislodged before they obtain a blood meal, but mosquitoes are able to obtain a blood meal more rapidly from a RVFV-infected animal than from an uninfected member of the same species. Again, the successful feeding rate on RVFV-infected animals would be higher than the percent of animals actually infected. Have either of these been incorporated in the models?

15. References: Please ensure that all references are formatted correctly.

 a. In manuscript titles, only the first word and all proper nouns should be capitalized. See references 1, 14 and others where “rift valley fever.” should have been “Rift Valley fever.” See also references 7, 8, 10, and others where each word is capitalized, even if not a proper noun. Please check all of the references.

 b. See references 41 and 46. Is it PLOS ONE or PLoS ONE?

 c. What does the “Larchmt N” mean in reference 54?

 d. Why do you list the editor for some of the PLoS journals?

 e. Please go through all of the references and ensure that they are properly formatted.

Minor comments:

16. Line 44: Both “Aedes” and “Culex” should be in italics.

17. Line 91: RVFV was established non line 57 and should be used here. See also lines 278, 290, 376, 466 etc.

18. Line 151, Figure 1: The first block indicates that 372 records were identified (236 + 136). The next block indicates that 123 of these were removed prior to screening. So why weren’t 249, instead of 248 screened? In all of the other reductions, the numbers do match?

19. Line 169: Shouldn’t “4 cases…” be “four cases…” See also lines 367, 391-394, 421, etc. I am fine with using a numeral within a parentheses, but you need to be consistent outside of a parentheses.

20. Line 342: Is it really necessary, or even accurate, to present lifetime to a hundredth of a day? Why not simply state, “under 8.7 days…”

21. Line 448: Why is 4/10 39% instead of 40%?

22. Line 480: Is it RVF infection or RVFV infection? This occurs throughout the manuscript. Mosquitoes do not transmit RVF, they transmit the virus that causes RVF.

23. Lines 499-500: Shouldn’t “on the lifespan of adult vectors lifespan” be “on the lifespan of adult vectors?”

24. Line 530: What is a SIR-type model? I assume that the authors are referring to a susceptible-infected-recovered model, but don’t know if the reader will get that. Also, in Table 1, is it “sensible” or “susceptible?” I think that susceptible is more accurate.

Reviewer's Responses to Questions

**Key Review Criteria Required for Acceptance?**

**Methods**

-Are the objectives of the study clearly articulated with a clear testable hypothesis stated?

-Is the study design appropriate to address the stated objectives?

-Is the population clearly described and appropriate for the hypothesis being tested?

-Is the sample size sufficient to ensure adequate power to address the hypothesis being tested?

-Were correct statistical analysis used to support conclusions?

-Are there concerns about ethical or regulatory requirements being met?

Reviewer #1: N/A - review paper, not hypothesis driven

y

y (but n/a hypothesis)

y (but n/a hypothesis)

N/A

N

Reviewer #2: -Are the objectives of the study clearly articulated with a clear testable hypothesis stated? NA

-Is the study design appropriate to address the stated objectives? Yes

-Is the population clearly described and appropriate for the hypothesis being tested? NA

-Is the sample size sufficient to ensure adequate power to address the hypothesis being tested? NA

-Were correct statistical analysis used to support conclusions? Yes

-Are there concerns about ethical or regulatory requirements being met? No

**Results**

-Does the analysis presented match the analysis plan?

-Are the results clearly and completely presented?

-Are the figures (Tables, Images) of sufficient quality for clarity?

Reviewer #1: y

y - very well presented

y - figures are excellent graphical communication of results

Reviewer #2: -Does the analysis presented match the analysis plan? Yes

-Are the results clearly and completely presented? Yes

-Are the figures (Tables, Images) of sufficient quality for clarity? Yes

**Conclusions**

-Are the conclusions supported by the data presented?

-Are the limitations of analysis clearly described?

-Do the authors discuss how these data can be helpful to advance our understanding of the topic under study?

-Is public health relevance addressed?

Reviewer #1: y

y

y

y

Reviewer #2: -Are the conclusions supported by the data presented? Yes

-Are the limitations of analysis clearly described? Yes

-Do the authors discuss how these data can be helpful to advance our understanding of the topic under study? Yes

-Is public health relevance addressed? NA

**Editorial and Data Presentation Modifications?**

Reviewer #1: Editorial recommendations:

- ms is very readable and accessible; authors do well bringing the reader into their view and objectives; good balance of technical and narrative content

- make sure each "e.g." and "i.e." is followed by a comma

- use serial commas (i.e., make sure there is a comma before the "and" or the "or" in a list of three or more items in text, such as line 118 or line 398)

- check that parenthetical statements are properly closed with a ")" ... e.g., in line 76. Overall, I would go through the whole ms to minimize use of parenthetical statements by rewording. However, the way you use them in for example lines 125-129 makes sense and works well.

- check for correct grammar with the word "data" which is a plural term - I think there was only one place (line 429) but check throughout

- be consistent with either numbering or using author/date in-text citations (e.g., in-text citations throughout the paragraph lines 365-376 are inconsistent with journal style)(on the other hand, the way you combine them in lines 159-171 makes sense and works well)

- check that "R" (as used in Ro, etc.) is italicized throughout

- lines 93-94: suggest rewording "To achieve this, we categorized models as either theoretical or applied, and explored these categories throughout the paper to identify what they have in common and how they differ."

- suggest find a way to make the count in line 31 and line 148 match (I understand it was 48 papers but 49 studies because one paper had two studies)

- lines 118-120: suggest rewording to "Records selected in the first and second step went to a full text screening of the corresponding report, using a combination of the first set of exclusion criteria along with these additional exclusion criteria:"

- line 238 should be four categories not five?

- lines 482-485: suggest rewording for clarity ... do you mean "Among 11 models including a human compartment (Table 1), three [42,64,73] considered human-to-mosquito transmission, 10 considered mosquito-to-human transmission, and one [67] considered livestock-to-human transmission."

Reviewer #2: Minor Revision

**Summary and General Comments**

Reviewer #1: First I want to thank you for a very well written, timely, and informative paper.

This manuscript synthesizes 48 Rift Valley fever virus modeling papers (49 studies) using very creative methods of analysis and communication of findings.

The figures are very well executed and valuable, and greatly enhance the text descriptions and tabulated analysis.

The ‘reading grid’ structured analysis of the papers makes a lot of sense and is a very defendable method for bringing together input from multiple co-authors.

The objective of the study is to summarize a state-of-the-science which also includes a gap analysis to guide future research.

The methodical, stepwise discussion of how key features of mechanistic disease models are handled across the studies will be very useful not only to guide future project designs, but also is very instructive and accessible for early career modelers.

I have a few recommendations to improve the ms for publication:

Lines 75-76 … can you elaborate on why phenomenological should be used instead of statistical?

Lines 93-95 … should you mention the “gray” category here?

Lines 98-99 … Force of Infection section (lines 524-621) … this section is very useful and interesting but possibly could be in the SI? I only say this because the intro paragraph (525-529) could almost skip straight to the “results” paragraph (starting 628) where you bring the narrative back to the papers you analyzed. As a reader, it left me a little lost until I understood what you were doing with the 524-621 section (i.e., setting the basis for your analysis and comment). Moving 524-621 to SI could keep the reader focused on your findings. Also, given that you devoted so much to the FOI issue, I expected more exploration and impact of this topic in the Conclusions.

Lines 247-263 (and possibly other sections, such as line 448) … the tallies you provide for the number of studies that attain various criteria throughout this paragraph are not identifiable by the reader to the specific studies. This is in contrast to other such tallies elsewhere in the ms that reference for example the Tables where the reader can translate each tally into the specific studies. You do reference Fig 3 but I don’t see a way to get to the specific studies tallied in 247-263 via this figure. This could be resolved by an additional table in SI that accompanies Fig. 3. Or maybe the existing tables have the relevant info and just need to be referenced in text?

Line 387 … could you list the countries?

Line 423 … is spatialized the same as data that are organically spatial? Or does it mean the data were fitted by the modeler to a spatial context?

Line 428 … could you describe what the specific data gaps were in those cited studies?

Line 468 … could add “Britch SC, Binepal YS, Ruder MG, Kariithi HM, Linthicum KJ, Anyamba A, et al. Rift Valley fever risk map model and seroprevalence in selected wild ungulates and camels from Kenya. PLoS One. 2013;8: e66626”

Line 480 … should be RVFV infection (not RVF infection)

Lines 485-487 … recommend citing Njenga et al. 2009 because humans can develop viremia sufficient to infect mosquitoes (Njenga MK, Paweska J, Wanjala R, Rao CY, Weiner M, Omballa V, et al. Using a field quantitative real-time PCR test to rapidly identify highly viremic Rift Valley fever cases. J Clin Microbiol. 2009;47: 1166–1171). Also Rolin et al. 2013 is a good reference for the idea of humans as dead end hosts (Rolin AI, Berrang-Ford L, Kulkarni MA. The risk of Rift Valley fever virus introduction and establishment in the United States and European Union. Emerg Microbes Infect. 2013;2: e81). We have recently submitted a review paper on the risk of spread of RVFV via infected humans - unfortunately still in review but will share with you if you are interested.

Lines 512-513 … the NDVI basis for modeling RVFV risk was pioneered by Anyamba and Linthicum … I realize their models may have not met your criteria for inclusion in this ms but the NDVI technique for understanding RVFV transmission risk should be cited to them (Linthicum KJ, Anyamba A, Tucker CJ, Kelley PW, Myers MF, Peters CJ. Climate and satellite indicators to forecast Rift Valley fever epidemics in Kenya. Science. 1999;285: 397–400; Anyamba A, Linthicum KJ, Mahoney R, Tucker CJ, Kelley PW. Mapping potential risk of Rift Valley fever outbreaks in African savannas using vegetation index time series data. Photogramm Eng Remote Sens. 2002;68: 137–145.) … in particular because the predictions of the Rift Valley Fever Monitor (https://www.ars.usda.gov/southeast-area/gainesville-fl/center-for-medical-agricultural-and-veterinary-entomology/docs/rvf_monthlyupdates/) guided mass vaccination and prevention of an epizootic (Linthicum KJ, Britch SC, Anyamba A. Rift Valley fever: An emerging mosquito-borne disease. Annu Rev Entomol. 2016;61: 395–415; Anyamba A, Chretien J-P, Britch SC, Soebiyanto RP, Small JL, Jepsen R, et al. Global Disease Outbreaks Associated with the 2015-2016 El Niño Event. Sci Rep. 2019;9: 1930.

Lines 659-660: yes absolutely … this is a very timely and operationally relevant conclusion and direction for future research. Movement (legal, illegal) of infected livestock among regions in RVFV endemic areas is becoming more and more important. Increasing instances of interepizootic transmission of RVFV to humans throughout the African continent and the Arabian Peninsula driven by movement of infected livestock (Eisa et al. 1980, Meegan and Bailey 1989, Balkhy and Memish 2003, LaBeaud et al. 2008, Mohamed et al. 2014, Napp et al. 2018, Tigoi et al. 2020) increase the risk of globalization of RVFV.

Balkhy HH, Memish ZA. Rift Valley fever: an uninvited zoonosis in the Arabian Peninsula. Int J Antimicrob Agents. 2003;21: 153–157.

Eisa M, Kheir el-Sid ED, Shomein AM, Meegan JM. An outbreak of Rift Valley fever in the Sudan - 1976. Trans R Soc Trop Med Hyg. 1980;74: 417–419.

LaBeaud AD, Muchiri EM, Ndzovu M, Mwanje MT, Muiruri S, Peters CJ, et al. Interepidemic Rift Valley fever virus seropositivity, northeastern Kenya. Emerg Infect Dis. 2008;14: 1240–1246.

Meegan JM, Bailey CL. Rift Valley fever. In: Monath TP, editor. The Arboviruses. CRC Press, Inc., Boca Raton, FL; 1989. pp. 51–76.

Mohamed AM, Ashshi AM, Asghar AH, Abd El-Rahim IHA, El-Shemi AG, Zafar T. Seroepidemiological survey on Rift Valley fever among small ruminants and their close human contacts in Makkah, Saudi Arabia, in 2011. Rev Sci Tech. 2014;33: 903–915.

Napp S, Chevalier V, Busquets N, Calistri P, Casal J, Attia M, et al. Understanding the legal trade of cattle and camels and the derived risk of Rift Valley fever introduction into and transmission within Egypt. PLoS Negl Trop Dis. 2018;12: e0006143.

Tigoi C, Sang R, Chepkorir E, Orindi B, Arum SO, Mulwa F, et al. High risk for human exposure to Rift Valley fever virus in communities living along livestock movement routes: A cross-sectional survey in Kenya. PLoS Negl Trop Dis. 2020;14: e0007979.

Reviewer #2: The review is quite detailed. The gap in literature was pointed out. Below are some comments and suggestions,

1. Page 2, Line 4, the sentence `... go the extra miles' should be rephreased.'

2. Page 2, Line 43, remove `in'.

3. Page 3, Line 73, the sentence needs to be rephrased.

4. Page 3, Line 83, change `risk' into `risks'.

5. Page 4, Line 93, remove `models in'.

6. Page 6, Line 139, change `model' into `models'.

7. Page 21, Line 333, change `was' into `were'.

8. Page 22, Line 342, change R0 ito $R_0$.

9. Page 22, Line 354, add comma after `However'.

10.Page 23, Line 371, remove `between'.

11. Page 31, Page 32, add commas between the two equalities in equations (1), (2), and (3).

PLOS authors have the option to publish the peer review history of their article (what does this mean?). If published, this will include your full peer review and any attached files.

Reviewer #1: Yes: Seth C. Britch

Reviewer #2: No
---

## [Decision Letter · Decision Letter 1]

16 Aug 2022

Dear Mr. Drouin,

Thank you very much for submitting a revised version of your manuscript "Mechanistic models of Rift Valley fever virus transmission dynamics: A systematic review" for consideration at PLOS Neglected Tropical Diseases. Based on the review, we are likely to accept this manuscript for publication, providing that you modify the manuscript according to my and the reviewer’s recommendations.

Below are a few more comments that need to be addressed.

Specific Commenrts:

1. Line 59: Yes, RVFV has been detected in several Anopheles species, but are Anopheles competent vectors of RVFV. Just because a mosquito feeds on a viremic animal and ingests virus does not make it a vector. Anopheles tend to have an extreme salivary gland barrier for RVFV and are virtually unable to transmit this virus. While rarely tested for RVFV, various species of sand flies are competent vectors, and RVFV is a member of the genus of sand fly fever viruses.

2. Line 195, Table 1: What does “Vector health status” mean? Should that be “Vector infection status?”

3. Line 283: Much better, but is “unrealistic” sufficient? Remember, The Romoser paper detected evidence of RVFV in one egg in 8.6% of the RVFV-inoculated mosquitoes. Because they were inoculated, this rate would be much higher than in mosquitoes that had ingested RVFV. Also, given that each mosquito lays about 150 eggs/oviposition cycle, the 8.6% of females with an infected egg translates to 0.06% of the progeny of the RVFV-inoculated mosquitoes would be infected. If the model used 20%, then to me, the model is beyond unrealistic when the rate would be <0.1% of the progeny of RVFV-inoculated mosquitoes. However, as you now point out, if a mosquito had a stabilized infection with RVFV, that would change everything.

4. Lines 304-306: Yes, the importation of an infectious disease into an area where it is not currently present can allow for a major outbreak. Why is that surprising? If the Ro is > 1, then you will get an outbreak. For RVF, I believe that the Ro is strongly affected by the presence (and number) of competent vectors and the presence of competent vertebrate amplifying hosts (e.g., cattle, goats, sheep, and various wildlife).

5. Lines 308-312: The association between bovine tuberculosis and RVF is interesting. Just curious, does infection with RVF make a cow more susceptible to BTB, or are herds that are in certain locations more like to be infected with both agents (i.e., neither pathogen affects the replication of the other, but certain areas are more likely to have either both or neither of the two pathogens)? Yes, if your model incorporates increases the amount of RVF with increasing amounts of BTB, isn’t it obvious that it will predict more RVFV in a population with a higher BTB positivity?

6. Lines 312-317: Again, better, but why even include a paper about ticks? Because RVFV is a member of the genus Phlebovirus (i.e., sand fly fever virus) and various species of sand flies have been proven (in the laboratory) to be competent vectors of RVFV, why hasn’t someone included them in the model, or in the case of this manuscript, why isn’t it pointed out that these models are missing? To me, that is much more important than devoting space to a tick model.

7. Lines 363-363: How the use of larvicides, even in a low transmission context could increase RVF incidences seems contra intuitive to me. How did this model find this increase in incidence? Yes, during an outbreak, larvicides would be ineffective as they would not reduce the number of infected vectors present. However, the use of adulticides would reduce the vector population and be more effective. So, if you compared areas that used larvicides with those that used adulticides, then there might be more transmission in the areas that used larvicides. However, that is not what the manuscript implies. 

8. Lines 379-383: Yes, I know that the model predicts that “stamping out in a 20km radius was the most effective strategy. However, given that this disease has never been observed in Europe, how accurate is that model? Would intensive mosquito control be critical. How many infectious cases in cattle would be subclinical and allow the virus to continue to spread? My concern is that given the lack of data going into these models, that a control practice based solely on this model may allow for RVFV to become enzootic.

9. Line 478: Yes, the models only applied to “livestock,” but is that really on a single vertebrate host species? Livestock consists of several species.

10. Line 512-516: Thanks for modifying this a little, but it illustrates a much bigger problem. Yes, many studies consider humans to be a dead-end host. This is probably because with nearly all of the other zoonotic arboviruses, i.e., eastern equine encephalitis virus, western equine encephalitis virus, Japanese encephalitis virus, etc., humans produce a very low viremia and are thus a dead-end host. Therefore, humans must be a dead end host for RVF too. However, studies indicate that the viremia in humans infected with RVFV can be extremely high, often >8 logs, i.e., higher than in adult cattle, goats, or sheep (see Meegan 1979, Trans R Soc Trop Med). Note that the de St. Maurice et al. 2017 paper cited (ref 139), only found viremias about 5-6 logs, but didn’t start testing until about 5 days after the onset of fever, by which time viremias would have fallen greatly. Yes, unfortunately, there are relatively few studies that have looked at viremias in humans, but even implying that humans are a dead-end host is misleading and dangerous. Given the number of humans that visit enzootic areas every year, if one of them became infected, given rapid transportation, they could easily transport RVFV back to an area (Europe or the Americas) where this virus is not yet present. Transportation of animals is important for local movement of RVF, but humans may be much more important for long range movement.

11. Lines 551-553: See also Rossignol PA, Ribeiro JM, Jungery M, Turell MJ, Spielman A, Bailey CL. Enhanced mosquito blood-finding success on parasitemic hosts: evidence for vector-parasite mutualism. Proc Natl Acad Sci U S A. 1985 Nov;82(22):7725-7. That paper showed that mosquitoes were able to feed significantly more rapidly on a RVFV-infected animal as compared to an uninfected one and is one of the explanations for why mosquitoes are more successful in feeding on a RVFV-infected animal. Not mentioned here is that infected animals may have less “anti-mosquito” behavior because they are ill or that because they are febrile that they might be more “attractive” to a host-seeking mosquito.

Minor: 

12. line 40: “modellers…” should be “modelers…”

13. Line 43: There should not be a “,” after “priorities” as the next phrase is not a complete sentence. Same comment on line 51 for the comma after “choices.”

14. Line 289 (and elsewhere): Is it “endemic” or “enzootic” circulation/areas?

15. Line 346: What do you mean by “both host populations?” Is it cattle and small ruminants or is it cattle and humans? Above the sentence seems to imply small ruminants, but the discussion below implies humans.

16. Line 394: Shouldn’t “Rift Valley fever virus transmission” be “RVFV transmission?

17. Line 452: Shouldn’t “exposure ; and” be “exposure; and” with no space before the colon?

18. Line 610: As “years” are a unit of measurement, it should be “5 years…” Similarly, on line 611, it should be “10 previous years…”

19. Line 640: Shouldn’t “Bibliography” be “References?”

20. References: Much better.

 a. Line 735, reference 40: “Rift Valley Fever…” should be “Rift Valley fever…?” See also references 76, 93, 125 and possibly others.

 b. Line 740, reference 42: Here, “Culex pipiens” is not in italics, but in the other references, species names have been in italics. Please be consistent.

 c. Line 1014, reference 145: Only the first word and proper nouns in a title should be capitalized.

Sincerely,

Michael J Turell, Ph.D.

Academic Editor

Robert Reiner

Section Editor

Thanks for the revised version and addressing many of the comments made by me and the reviewers. Below are a few more comments that need to be addressed.

Specific Commenrts:

1. Line 59: Yes, RVFV has been detected in several Anopheles species, but are Anopheles competent vectors of RVFV. Just because a mosquito feeds on a viremic animal and ingests virus does not make it a vector. Anopheles tend to have an extreme salivary gland barrier for RVFV and are virtually unable to transmit this virus. While rarely tested for RVFV, various species of sand flies are competent vectors, and RVFV is a member of the genus of sand fly fever viruses.

2. Line 195, Table 1: What does “Vector health status” mean? Should that be “Vector infection status?”

3. Line 283: Much better, but is “unrealistic” sufficient? Remember, The Romoser paper detected evidence of RVFV in one egg in 8.6% of the RVFV-inoculated mosquitoes. Because they were inoculated, this rate would be much higher than in mosquitoes that had ingested RVFV. Also, given that each mosquito lays about 150 eggs/oviposition cycle, the 8.6% of females with an infected egg translates to 0.06% of the progeny of the RVFV-inoculated mosquitoes would be infected. If the model used 20%, then to me, the model is beyond unrealistic when the rate would be <0.1% of the progeny of RVFV-inoculated mosquitoes. However, as you now point out, if a mosquito had a stabilized infection with RVFV, that would change everything.

4. Lines 304-306: Yes, the importation of an infectious disease into an area where it is not currently present can allow for a major outbreak. Why is that surprising? If the Ro is > 1, then you will get an outbreak. For RVF, I believe that the Ro is strongly affected by the presence (and number) of competent vectors and the presence of competent vertebrate amplifying hosts (e.g., cattle, goats, sheep, and various wildlife).

5. Lines 308-312: The association between bovine tuberculosis and RVF is interesting. Just curious, does infection with RVF make a cow more susceptible to BTB, or are herds that are in certain locations more like to be infected with both agents (i.e., neither pathogen affects the replication of the other, but certain areas are more likely to have either both or neither of the two pathogens)? Yes, if your model incorporates increases the amount of RVF with increasing amounts of BTB, isn’t it obvious that it will predict more RVFV in a population with a higher BTB positivity?

6. Lines 312-317: Again, better, but why even include a paper about ticks? Because RVFV is a member of the genus Phlebovirus (i.e., sand fly fever virus) and various species of sand flies have been proven (in the laboratory) to be competent vectors of RVFV, why hasn’t someone included them in the model, or in the case of this manuscript, why isn’t it pointed out that these models are missing? To me, that is much more important than devoting space to a tick model.

7. Lines 363-363: How the use of larvicides, even in a low transmission context could increase RVF incidences seems contra intuitive to me. How did this model find this increase in incidence? Yes, during an outbreak, larvicides would be ineffective as they would not reduce the number of infected vectors present. However, the use of adulticides would reduce the vector population and be more effective. So, if you compared areas that used larvicides with those that used adulticides, then there might be more transmission in the areas that used larvicides. However, that is not what the manuscript implies. 

8. Lines 379-383: Yes, I know that the model predicts that “stamping out in a 20km radius was the most effective strategy. However, given that this disease has never been observed in Europe, how accurate is that model? Would intensive mosquito control be critical. How many infectious cases in cattle would be subclinical and allow the virus to continue to spread? My concern is that given the lack of data going into these models, that a control practice based solely on this model may allow for RVFV to become enzootic.

9. Line 478: Yes, the models only applied to “livestock,” but is that really on a single vertebrate host species? Livestock consists of several species.

10. Line 512-516: Thanks for modifying this a little, but it illustrates a much bigger problem. Yes, many studies consider humans to be a dead-end host. This is probably because with nearly all of the other zoonotic arboviruses, i.e., eastern equine encephalitis virus, western equine encephalitis virus, Japanese encephalitis virus, etc., humans produce a very low viremia and are thus a dead-end host. Therefore, humans must be a dead end host for RVF too. However, studies indicate that the viremia in humans infected with RVFV can be extremely high, often >8 logs, i.e., higher than in adult cattle, goats, or sheep (see Meegan 1979, Trans R Soc Trop Med). Note that the de St. Maurice et al. 2017 paper cited (ref 139), only found viremias about 5-6 logs, but didn’t start testing until about 5 days after the onset of fever, by which time viremias would have fallen greatly. Yes, unfortunately, there are relatively few studies that have looked at viremias in humans, but even implying that humans are a dead-end host is misleading and dangerous. Given the number of humans that visit enzootic areas every year, if one of them became infected, given rapid transportation, they could easily transport RVFV back to an area (Europe or the Americas) where this virus is not yet present. Transportation of animals is important for local movement of RVF, but humans may be much more important for long range movement.

11. Lines 551-553: See also Rossignol PA, Ribeiro JM, Jungery M, Turell MJ, Spielman A, Bailey CL. Enhanced mosquito blood-finding success on parasitemic hosts: evidence for vector-parasite mutualism. Proc Natl Acad Sci U S A. 1985 Nov;82(22):7725-7. That paper showed that mosquitoes were able to feed significantly more rapidly on a RVFV-infected animal as compared to an uninfected one and is one of the explanations for why mosquitoes are more successful in feeding on a RVFV-infected animal. Not mentioned here is that infected animals may have less “anti-mosquito” behavior because they are ill or that because they are febrile that they might be more “attractive” to a host-seeking mosquito.

Minor: 

12. line 40: “modellers…” should be “modelers…”

13. Line 43: There should not be a “,” after “priorities” as the next phrase is not a complete sentence. Same comment on line 51 for the comma after “choices.”

14. Line 289 (and elsewhere): Is it “endemic” or “enzootic” circulation/areas?

15. Line 346: What do you mean by “both host populations?” Is it cattle and small ruminants or is it cattle and humans? Above the sentence seems to imply small ruminants, but the discussion below implies humans.

16. Line 394: Shouldn’t “Rift Valley fever virus transmission” be “RVFV transmission?

17. Line 452: Shouldn’t “exposure ; and” be “exposure; and” with no space before the colon?

18. Line 610: As “years” are a unit of measurement, it should be “5 years…” Similarly, on line 611, it should be “10 previous years…”

19. Line 640: Shouldn’t “Bibliography” be “References?”

20. References: Much better.

 a. Line 735, reference 40: “Rift Valley Fever…” should be “Rift Valley fever…?” See also references 76, 93, 125 and possibly others.

 b. Line 740, reference 42: Here, “Culex pipiens” is not in italics, but in the other references, species names have been in italics. Please be consistent.

 c. Line 1014, reference 145: Only the first word and proper nouns in a title should be capitalized.

Reviewer's Responses to Questions

**Key Review Criteria Required for Acceptance?**

**Methods**

-Are the objectives of the study clearly articulated with a clear testable hypothesis stated?

-Is the study design appropriate to address the stated objectives?

-Is the population clearly described and appropriate for the hypothesis being tested?

-Is the sample size sufficient to ensure adequate power to address the hypothesis being tested?

-Were correct statistical analysis used to support conclusions?

-Are there concerns about ethical or regulatory requirements being met?

Reviewer #3: (No Response)

**Results**

-Does the analysis presented match the analysis plan?

-Are the results clearly and completely presented?

-Are the figures (Tables, Images) of sufficient quality for clarity?

Reviewer #3: (No Response)

**Conclusions**

-Are the conclusions supported by the data presented?

-Are the limitations of analysis clearly described?

-Do the authors discuss how these data can be helpful to advance our understanding of the topic under study?

-Is public health relevance addressed?

Reviewer #3: (No Response)

**Editorial and Data Presentation Modifications?**

Reviewer #3: (No Response)

**Summary and General Comments**

Reviewer #3: This study represents a valuable exercise in reviewing all modeling studies of RVFV using a systematic literature review. Although reviews of the RVFV system have occurred frequently, this study updates a synthesis of published models and provides an excellent framework for evaluating different aspects of past applied and theoretical models. The inclusion of inheritance connections among the studies is also a nice feature. I have a few minor observations to improve the manuscript.

Ln. 144. Could ‘and insights remain distant from field preoccupations’ be explained better as I don’t understand what that means.

Ln. 417-475. This is a great observation to point out. However, every author on this current systematic review appears to have an affiliation with an institution in France.

Ln. 518. This statement ‘while models with one taxon often did not precise the genus of vector studied’ could be changed to ‘while models with one vector taxon did not specify genus or species’ for clarity.

Figure 2 and 5. The legend nor text explain what ‘FR’, ‘FI’, mean in either of these figures. The Box explains but it would be helpful to at least include the name of the abbreviation to help interpret the figures. 

Figure 4. For the ‘Location and number of RVF models.’. Does this ‘location’ refer to where the authors were from or where the model was parameterized in terms of geographic location? I assume it is the latter but it is hard to tell this in the methods or this Figure 4 legend so this could be clarified.

PLOS authors have the option to publish the peer review history of their article (what does this mean?). If published, this will include your full peer review and any attached files.

Reviewer #3: No

Figure Files:

Data Requirements:

Reproducibility:

References

---

## [Editor Report · Decision Letter 2]

23 Sep 2022

Dear Mr. Drouin,

Thank you very much for submitting a revised version of your manuscript, "Mechanistic models of Rift Valley fever virus transmission: A systematic review" for consideration at PLOS Neglected Tropical Diseases. I appreciated the attention to an important topic. We are likely to accept this manuscript for publication, providing that you modify the manuscript according to the comments below.

Thanks for the revised version and addressing the comments made by me and the reviewer. My only real issue is stating that people consider humans to be a dead-end host for RVFV. Please see my comment 5 below as well as a few minor comments. 

1. Line 33: Insert a space between “models” and “with”

2. Line 39: Insert a space after the period after “transmission” and shouldn’t this be in the past tense, so shouldn’t “we note a…” be “we noted a…?”

3. Line 193: Shouldn’t there be a “period” after “et al” instead of a “comma?”

4. Lines 370-372: Again, how could the use of larvicides increase the proportion of infected mosquitoes? Yes, the use of larvicides is generally ineffective in disease prevention. This is because it does not kill any of the infectious mosquitoes that are already out there. Was this only in their model, or is there any real-world evidence that the use of larvicides led to an increased evidence of disease.

5. Lines 519-523: Yes, there are a number of papers that have mentioned that humans are a dead-end host for RVFV. Note, if this paper is publish as currently written, it too will be cited as evidence that humans are a dead-end host for RVFV. They are not! Unfortunately, as humans are dead-end hosts for most zoonotic viruses (West Nile, eastern equine encephalitis, Japanese encephalitis, St. Louis encephalitis, etc.) some people assumed that humans must be a dead-end host of this zoonotic pathogen. In the very few studies that have looked at viremias in active human infections, viremias can be very high (actually higher than in cattle or sheep). That being said, humans probably play a very small role in the outbreak spread of RVFV because the reduced number of mosquito bites/day on a human compared to a cow. However, that small role is not due to them being a dead-end host, which implies that the viremia is too low to infect a mosquito. In fact, humans are probably the most likely way that RVFV will eventually spread to the Americas and Europe, i.e., in a tourist bitten by an infectious mosquito while visiting one of the enzootic areas to look at wildlife. I do not want this paper to be another one cited that humans are a dead-end host. How about, “…human-to-mosquito transmission. This is in line with humans not being considered important in the outbreak spread of RVFV because of limited mosquito biting of humans compared with livestock. However, some…” 

6. Line 550: Should “genus” be “genera” as it could be more than one genus?

Sincerely,

Michael J Turell, Ph.D.

Academic Editor

Robert Reiner

Section Editor

Thanks for the revised version and addressing the comments made by me and the reviewer. My only real issue is stating that people consider humans to be a dead-end host for RVFV. Please see my comment 5 below as well as a few minor comments. 

1. Line 33: Insert a space between “models” and “with”

2. Line 39: Insert a space after the period after “transmission” and shouldn’t this be in the past tense, so shouldn’t “we note a…” be “we noted a…?”

3. Line 193: Shouldn’t there be a “period” after “et al” instead of a “comma?”

4. Lines 370-372: Again, how could the use of larvicides increase the proportion of infected mosquitoes? Yes, the use of larvicides is generally ineffective in disease prevention. This is because it does not kill any of the infectious mosquitoes that are already out there. Was this only in their model, or is there any real-world evidence that the use of larvicides led to an increased evidence of disease.

5. Lines 519-523: Yes, there are a number of papers that have mentioned that humans are a dead-end host for RVFV. Note, if this paper is publish as currently written, it too will be cited as evidence that humans are a dead-end host for RVFV. They are not! Unfortunately, as humans are dead-end hosts for most zoonotic viruses (West Nile, eastern equine encephalitis, Japanese encephalitis, St. Louis encephalitis, etc.) some people assumed that humans must be a dead-end host of this zoonotic pathogen. In the very few studies that have looked at viremias in active human infections, viremias can be very high (actually higher than in cattle or sheep). That being said, humans probably play a very small role in the outbreak spread of RVFV because the reduced number of mosquito bites/day on a human compared to a cow. However, that small role is not due to them being a dead-end host, which implies that the viremia is too low to infect a mosquito. In fact, humans are probably the most likely way that RVFV will eventually spread to the Americas and Europe, i.e., in a tourist bitten by an infectious mosquito while visiting one of the enzootic areas to look at wildlife. I do not want this paper to be another one cited that humans are a dead-end host. How about, “…human-to-mosquito transmission. This is in line with humans not being considered important in the outbreak spread of RVFV because of limited mosquito biting of humans compared with livestock. However, some…” 

6. Line 550: Should “genus” be “genera” as it could be more than one genus?

Figure Files:

Data Requirements:

Reproducibility:

References

---

## [Editor Report · Decision Letter 3]

31 Oct 2022

Dear Mr. Drouin,

Thank you for making the suggested changes and we are pleased to inform you that your manuscript 'Mechanistic models of Rift Valley fever virus transmission: A systematic review' has been provisionally accepted for publication in PLOS Neglected Tropical Diseases.

Best regards,

Michael J Turell, Ph.D.

Academic Editor

Robert Reiner

Section Editor

---

## [Editor Report · Acceptance letter]

9 Nov 2022

Dear Dr. Durand,

We are delighted to inform you that your manuscript, "Mechanistic models of Rift Valley fever virus transmission: A systematic review," has been formally accepted for publication in PLOS Neglected Tropical Diseases.

Best regards,

Shaden Kamhawi

co-Editor-in-Chief

Paul Brindley

co-Editor-in-Chief
